# Dysregulation of cystathionine γ-lyase promotes prostate cancer progression and metastasis

Yi-Hsiang Wang[1,2,†], Jo-Ting Huang[1,†], Wen-Ling Chen[1], Rong-Hsuan Wang[3], Ming-Chien Kao[3], Yan-Ru Pan[3], Shih-Hsuan Chan[1,2,4], Kuo-Wang Tsai[5,6,7], Hsing-Jien Kung[1,8], Kai-Ti Lin[1,3,*] (iD) & Lu-Hai Wang[1,4,**] (iD)

## Abstract

Hydrogen sulfide (H$_2$S), an endogenous signaling gaseous molecule, is involved in various physiological activities, including vessel relaxation, regulation of cellular bioenergetics, inflammation, and angiogenesis. By using xenograft orthotopic implantation of prostate cancer PC3 cells and subsequently comparing bone metastatic with primary tumor-derived cancer cells, we find that H$_2$S-producing enzyme cystathionine γ-lyase (CTH) is upregulated in bone-metastatic PC3 cells. Clinical data further reveal that the expression of CTH is elevated in late-stage prostate cancer patients, and higher CTH expression correlates with poor survival from The Cancer Genome Atlas (TCGA) prostate cancer RNA-seq datasets. CTH promotes NF-κB nuclear translocation through H$_2$S-mediated sulfhydration on cysteine-38 of the NF-κB p65 subunit, resulting in increased IL-1β expression and H$_2$S-induced cell invasion. Knockdown of CTH in PC3 cells results in the suppression of tumor growth and distant metastasis, while overexpression of CTH in DU145 cells promotes primary tumor growth and lymph node metastasis in the orthotopic implanted xenograft mouse model. Together, our findings provide evidence that CTH generated H$_2$S promotes prostate cancer progression and metastasis through IL-1β/NF-κB signaling pathways.

**Keywords** cystathionine gamma-lyase; hydrogen sulfide; IL-1beta; NF-kappaB; prostate cancer

**Subject Categories** Cancer; Post-translational Modifications & Proteolysis; Signal Transduction

## Introduction

Prostate cancer (PC) is the most common male malignancy in the United States [1] and the sixth leading cause of cancer deaths in men worldwide [2]. Most deaths from PC are caused by metastases, especially in bone [3]. Bone metastasis is virtually incurable and can lead to pain, pathological fractures, nerve compression syndromes, and hypercalcemia. Current treatments for bone-metastatic PC are mainly palliative, and the underlying mechanism leading to the bone metastasis of PC remains unclear. Novel therapeutic strategy against bone metastasis of PC is urgently required to improve patient outcomes.

Hydrogen sulfide (H$_2$S), a rotten egg-smelling gas, is a signaling molecule that acts as a critical mediator in multiple physiological processes, including regulation of blood vessel vasodilation [4–6], cardiac response to ischemia/reperfusion injury [7], and inflammation [8]. H$_2$S modulates diverse cellular processes mainly by modifying specific cysteine residues in proteins through the formation of a persulfide (-SSH) bond; this modification is called protein sulfhydration [9].

H$_2$S is produced endogenously as a metabolite with cystathionine β-synthase (CBS), cystathionine γ-lyase (CTH), and 3-mercaptopyruvate sulfurtransferase (3MST). CBS catalyzes H$_2$S by a beta-replacement reaction with cysteine [10], while CTH produces H$_2$S from cysteine and water through alpha and beta elimination. In higher concentration of homocysteine, CTH also generates homolanthionine and H$_2$S from two homocysteines by gamma-replacement reactions [11]. Other than CBS and CTH, 3MST produces H$_2$S by processing 3-mercaptopyruvate (3MP), a metabolite catalyzed from L- and D-cysteine by cysteine aminotransferase (CAT) and D-amino acid oxidase (DAO), respectively [12–15]. Endogenous H$_2$S derived from these three enzymes plays an influential role on tumor growth in a variety of different cancer types through induction of

1   Institute of Molecular and Genomic Medicine, National Health Research Institutes, Zhunan, Miaoli County, Taiwan
2   Institute of Molecular Medicine, College of Life Science, National Tsing Hua University, Hsinchu, Taiwan
3   Institute of Biotechnology, College of Life Science, National Tsing Hua University, Hsinchu, Taiwan
4   Chiese Medicine Research Center, Institute of Integrated Medicine, China Medical University, Taichung City, Taiwan
5   Department of Medical Education and Research, Kaohsiung Veterans General Hospital, Kaohsiung, Taiwan
6   Institute of Biomedical Sciences, National Sun Yat-Sen University, Kaohsiung, Taiwan
7   Department of Chemical Biology, National Pingtung University of Education, Pingtung, Taiwan
8   PhD Program for Cancer Biology and Drug Discovery, Taipei Medical University, Taipei, Taiwan
    *Corresponding author. Tel: +88 6351 62592; E-mail: ktlin@life.nthu.edu.tw
    **Corresponding author. Tel: +88 6422 053366 #1012; E-mail: luhaiwang@mail.cmu.edu.tw
    †These authors contributed equally to this work

angiogenesis, regulation of mitochondrial bioenergetics, acceleration of cell cycle progression, and anti-apoptosis [16]. Among them, CBS was upregulated in thyroid [17], ovarian [18], and colon cancers [19,20], while CTH promoted cell growth in breast cancer cells [21] and astrocytoma cells [22]. Increased expressions of 3MST were observed in colon, lung adenocarcinoma, urothelial cell carcinoma, and oral carcinomas, implying that 3MST may play a role in drug resistance and oxidative damage during cancer progression [23]. Notably, it remains unclear whether $H_2S$ affects cancer metastasis.

Nuclear factor-kappaB (NF-κB), initially discovered and studied as a major transcription factor of immune and inflammatory functions, contributes in the control of cell proliferation and oncogenesis [24]. Studies indicate that activation of NF-κB signaling positively correlates with PC progression, chemoresistance [25], PSA recurrence [26,27], and metastatic spread [28,29]. The NF-κB activation gene signature effectively predicted metastasis-free survival in patients with PC [30], suggesting that genes downstream from NF-κB may play important roles in promoting PC metastasis. Among them, interleukin (IL)-1β prompted PC cells to grow into large skeletal lesions in mice [31] and to mediate cross-talk between osteoblasts and PC cells [32]. These observations indicate a potential role of IL-1β in the bone-metastatic progression of PC.

The post-translational modifications of NF-κB determine the duration and strength of NF-κB nuclear activity and transcriptional output [33]. Recent studies revealed that TNF-α stimulated the transcription of CTH and $H_2S$ subsequently generated sulfhydration on cysteine-38 of the p65 subunit of NF-κB, resulting in the transcriptional activation of NF-κB and expression of several anti-apoptotic genes [34]. This sulfhydration modification on the p65 subunit appears to be an important mechanism in modulating NF-κB transcriptional activities.

In the present study, we investigated the role of CTH and its derivative product, $H_2S$, in PC progression and metastasis. Our study showed that the expression of CTH increased the $H_2S$ level, resulting in the activation of NF-κB-mediated IL-1β signaling to instigate cell invasion by sulfhydration on cysteine-38 of the NF-κB p65 subunit. The orthotopic implantation study showed that knockdown of CTH in PC3 cells resulted in the suppression of primary tumor growth and lower incidence of lymph node and bone metastasis, while overexpression of CTH in DU145 cells promoted primary tumor growth and increased incidence of lymph node metastasis. Moreover, increased expression of CTH correlated with PC progression and poor survival of the patients. Overall, our current study identified CTH and its derivative $H_2S$ as potential therapeutic modalities in the intervention of PC progression and distant metastasis.

# Results

### Expression of CTH was upregulated in bone-metastatic PC cells

To elucidate the differential expression of genes during PC metastasis, we established paired PC3-derived cell lines isolated and established from tumor cells in bone marrow and from primary tumors, respectively, using xenograft orthotopic implantation mouse model [35]. Briefly, PC3 cells were injected into the ventral portion of the prostate in nude mice and allowed 40 days for the growth of the primary tumor and potential distant metastasis. We then cultured

tumor cells isolated from the primary prostate tumors as the "tumor-derived PC3 cells" or T lines, as well as tumor cells from thigh bone marrows to obtain the "bone-metastatic PC3 cells" or B lines. Seven paired PC3 B and T lines were thus established from bone marrows and primary tumors, respectively, from seven PC3 tumor-bearing nude mice. To identify genes involved in prostate cancer metastasis, we subsequently compared expression profiles between these paired cell lines. We found an enzyme, cystathionine γ-lyase (CTH), was upregulated in five of seven pairs of bone-metastatic PC3 cells from microarray analysis (Fig EV1A), and protein expression of CTH among these pairs was validated by Western blot analysis (Fig EV1B). To explore the biological role of CTH in PC, we examined the CTH expression levels in different PC cell lines (Fig EV1C). Consistent with previous studies [36], the CTH level was absent in the prostate epithelial cell line, RWPE-1, but was detectable in all other PC cell lines tested, including LNCaP, C4-2, PC3, 22Rv1, and DU-145.

To explore the functional relevance of CTH in PC, two pairs of bone-metastatic (PC3-B2 or PC3-B3) versus primary tumor-derived (PC3-T2 or PC3-T3) PC3 cells were selected. We observed that bone-metastatic PC3 cells exhibited substantially higher migration and invasion abilities (Fig 1A), but the cell proliferation rate remained similar between the B and T lines (Fig 1B). The expression of CTH and its enzymatic gaseous product, hydrogen sulfide ($H_2S$), was increased in PC3-B2 and PC3-B3 cells, as compared with PC3-T2 and PC3-T3 cells (Fig 1C and D). More importantly, to determine whether B lines exhibited higher metastatic potential than T lines, we re-inoculated PC3-B2, PC3-B3, and PC3-T2 and PC3-T3 cells orthotopically into the prostates in nude mice. Our data indicated that PC3-B2 and PC3-B3 exhibited higher metastatic potential than PC3-T2 and PC3-T3 cells (Fig 1E and F, Appendix Fig S1), indicating that PC3-B2 and PC3-B3 cells gained increased migratory ability to disseminate to distant organs such as paraaortic lymph nodes and bone marrow and grew into micro- or macrometastatic tumors, whereas PC3-T2 and PC3-T3 cells had much less ability to do so. We further examined protein expressions of the other two $H_2S$ producing enzymes, CBS and 3MST, in PC3-B2 and PC3-B3 cells. In contrast to the results with CTH, the expressions of CBS and 3MST remained similar in PC3-B2 and PC3-B3 cells (Fig EV1D). Since CTH is a key enzyme for glutathione (GSH) biosynthesis, we also checked whether the oxidative stress by GSH increased in PC3-B lines. Both total GSH levels and the glutathione disulfide (GSSG)/GSH ratio remained unchanged in PC3-B2 and PC3-B3 cells, as compared with PC3-T2 and PC3-T3 cells (Fig EV1E and F), suggesting that redox homeostasis is not affected in bone-metastatic PC3 cells.

### Increased expression of CTH correlated with progression and poor survival in PC

To examine the expression of CTH during PC progression, we evaluated CTH expression in PC specimens with defined stages and grades using immunohistochemistry and a tumor TNM staging system for grouping. The evaluations included 105 primary prostate tumors and 11 adjacent non-tumors. The staining intensities correlated well with the PC stage (Fig 1G). The expression of CTH in primary tumors increased considerably in patients with advanced stages (III/IV), as compared with the adjacent non-tumor tissue or

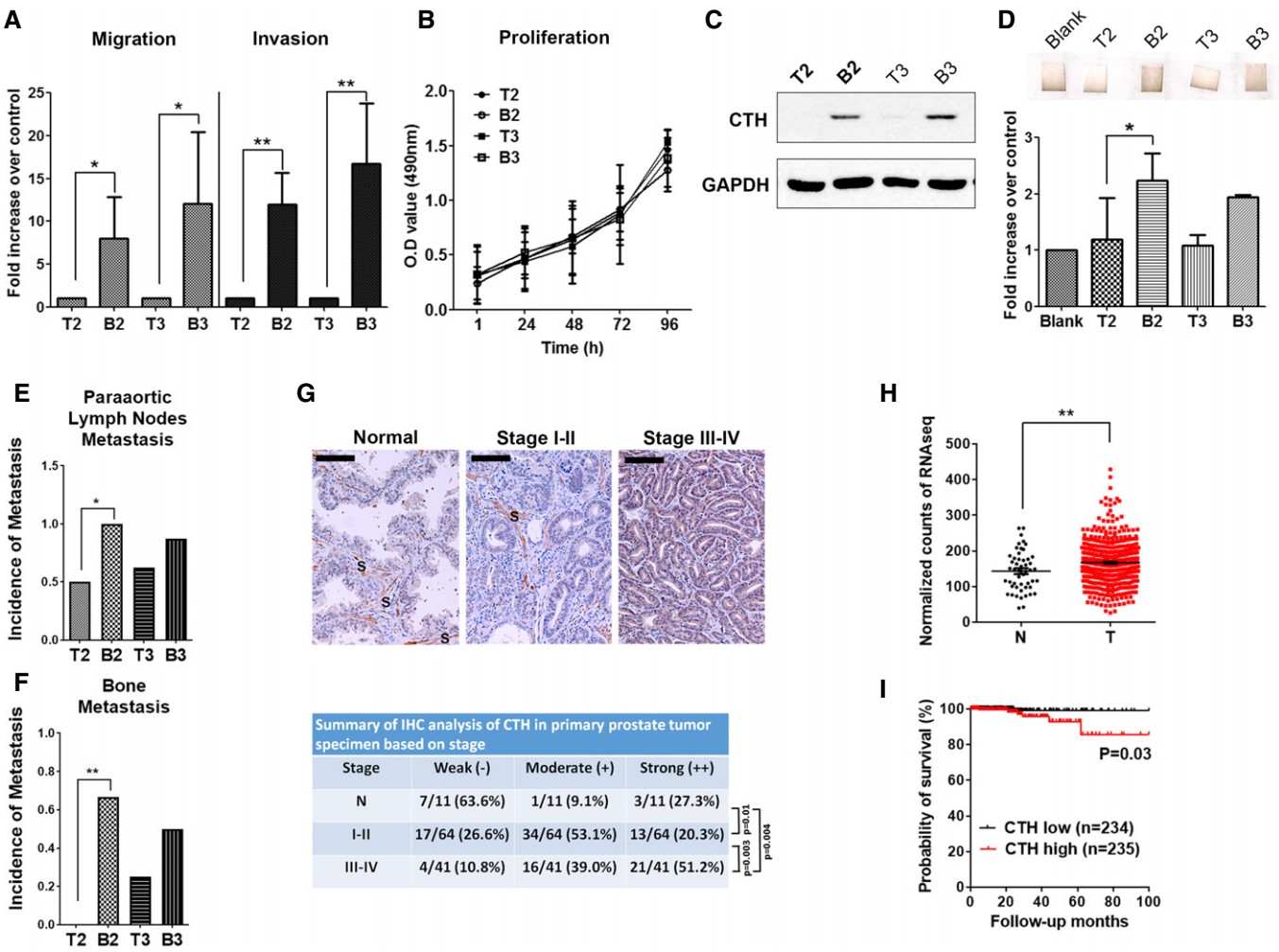

**Figure 1. Expression of CTH increases in bone-metastatic PC3 cells and positively correlates with PC progression and poor survival.**

A   Comparison of migration and invasion abilities between T and B lines. PC3 T and B line cells were incubated for migration (16 h) and invasion (24 h) assay. Data shown represent the means ± SD (*n* = 3 biological replicates). Student's *t*-test was used for the statistical analysis (\*P < 0.05; \*\*P < 0.01).

B   Comparison of cell proliferation rate between T and B lines. Cell proliferation assay was performed by MTS reagents. Data are the means ± SD (*n* = 3 biological replicates).

C   Western blot analysis of the endogenous CTH in PC3 orthotopically implanted primary tumor-derived (T2 and T3 lines) and bone metastasis-derived cancer cells (B2 and B3 lines).

D   Top: $H_2S$ catalyzing activity of cell lysates from PC3-T2, PC3-B2, PC3-T3, and PC3-B3 lines. Lead acetate-soaked paper strips show a PbS brown stain as a result of reaction with $H_2S$. Bottom: The level of $H_2S$ production was quantified by densitometry, and the histograms represent the means ± SD (*n* = 3 biological replicates). ANOVA followed by Tukey's post hoc test was used for the statistical analysis (\*P < 0.05).

E, F   $1 \times 10^6$ PC3-T2, PC3-B2, PC3-T3, and PC3-B3 cells were orthotopically injected into mouse prostate for 15–20 days. Incidences of paraaortic lymph nodes metastasis (E) and bone metastasis (F) are shown (*n* = 6–8 mice per group). Student's *t*-test was used for the statistical analysis (\*P < 0.05; \*\*P < 0.01).

G   Top: The expression of CTH in prostate tumors by immunocytochemistry analysis of commercial tissue arrays. Samples were sub-grouped by TNM stages. The representative images from different stages are shown. S: stroma; Scale bars: 250 μm. Bottom: The statistical significance was determined using the chi-square test

H, I   The RNA-seq data of CTH were downloaded from the TCGA database as described in the section of Materials and Methods. (H) The mRNA expression of CTH in PC specimens and normal tissues. Data are presented as means ± SD (*n* = 50 cases for adjacent non-tumor parts and 469 cases for PC tumors). Student's *t*-test was used for the statistical analysis (\*\*P < 0.01). (I) Kaplan–Meier survival analysis of PC patients with high or low CTH expression. The statistical significance was determined using the chi-square test.

Source data are available online for this figure.

early-stage tumors (I/II; Fig 1G). To explore the clinical significance of CTH, we analyzed the correlation of CTH expression with overall survival from the TCGA PC RNA-seq datasets. A total of 469 PC and 50 normal tissue specimens were included. Our analysis revealed that CTH expression substantially increased in PC tissues, as compared with the normal tissues (Fig 1H). More importantly,

analysis from PC patients with 8 years of follow-up revealed a significant correlation between high CTH expression and poor survival (Fig 1I), indicating the prognostic potential of CTH in PC. Concordantly, the association between CTH expression and poor survival was also observed in pancreatic adenocarcinoma (PAAD) and lower grade glioma (LGG) from the TCGA RNA-seq datasets

(Fig EV1G and H), implying that CTH may play a crucial role during cancer progression in various cancer types.

## CTH promoted PC cell migration and invasion, but not proliferation

We next investigated the role of CTH in PC progression. Overexpression of CTH from the pCMV-CTH-HA expression plasmid in PC3-T2 cells resulted in the increased $H_2S$ production, whereas depletion of CTH by siRNA in PC3-B2 cells reduced the production of $H_2S$ (Fig 2A and B). Knockdown of CTH significantly suppressed cell migration and invasion in PC3-B2 cells (Fig 2C). To ensure the specificity of the siRNA, we further confirmed that depletion of CTH by two different siRNAs significantly suppressed cell migration and

invasion in PC3-B2 and PC3-B3 cells, while re-expression of CTH successfully rescued reduced cell migration and invasion upon knockdown of CTH (Fig EV2A–C). Consistent with Fig 1B, the CTH knockdown did not affect the growth rates of PC3-B2 cells versus those of PC3-T2 cells (Fig 2D). Both migration and invasion abilities were significantly increased in the PC3 cells overexpressing CTH (Fig 2E and F), implying a potential role of CTH in cell migration/invasion. Knockdown of CTH in other PC cells lines, including 22Rv1 and C4-2, also resulted in the suppression of both cell migration and invasion (Fig EV2D and E), indicating the role of CTH in cell migration and invasion is common among the PC cell lines tested. Treatment of DL-propargylglycine (PAG), an irreversible inhibitor for CTH enzyme activity, resulted in the considerably reduced invasion but had no effect on cell migration (Fig 2G),

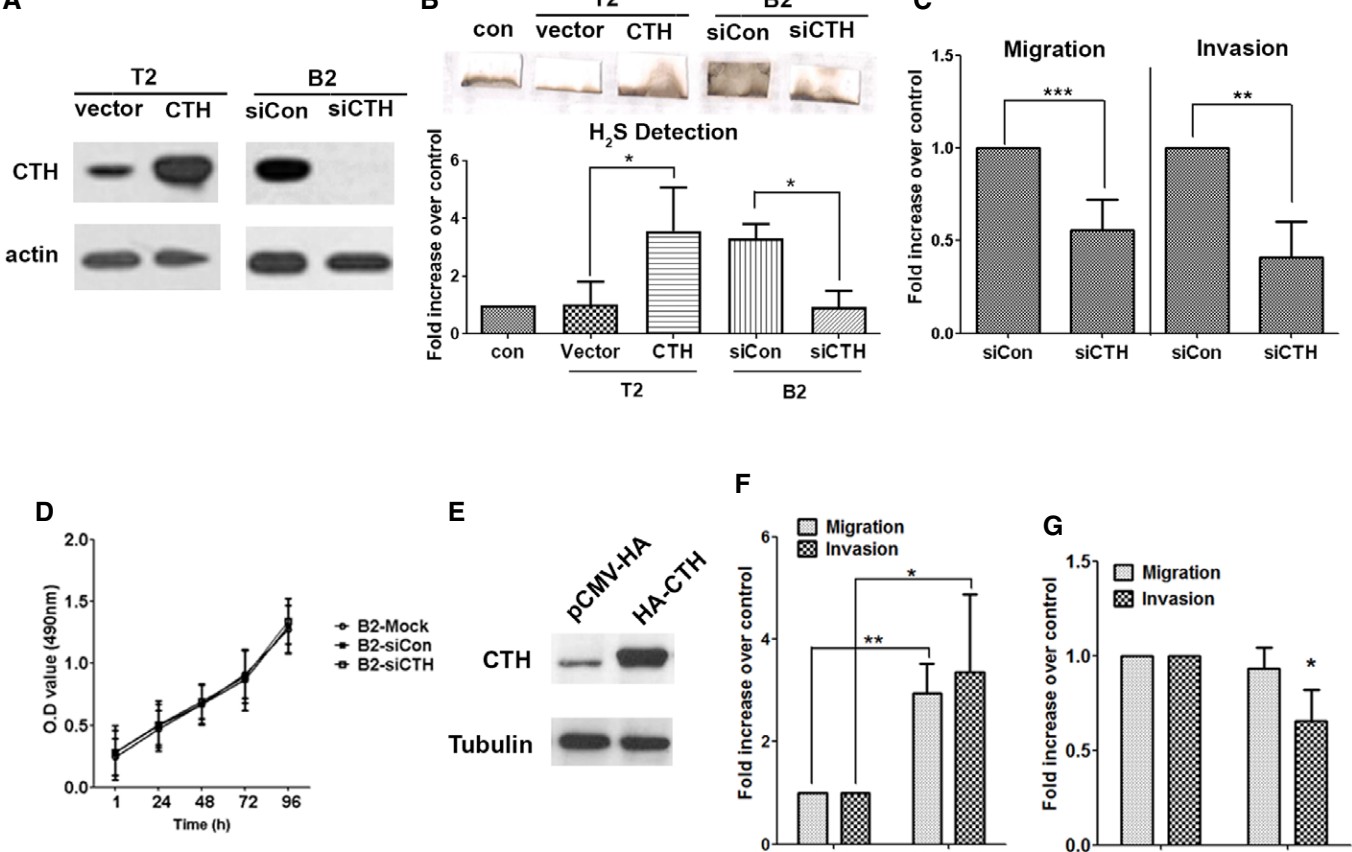

**Figure 2. CTH promotes $H_2S$ production, migration, and invasion.**

A Western blot analysis of the CTH level in PC3-T2 cells with CTH overexpression by pCMV-CTH-HA or PC3-B2 cells with CTH knockdown by siCTH-1.

B Top: The $H_2S$ production capacity of cell lysates from PC3-T2 cells with CTH overexpression by pCMV-CTH-HA or PC3-B2 cells with CTH knockdown by siCTH-1. Lead acetate-soaked paper strips show a PbS brown stain as a result of reaction with $H_2S$. Bottom: The level of $H_2S$ production was quantified by densitometry, and the histograms represent the means ± SD ($n$ = 3 biological replicates). ANOVA followed by Tukey's post hoc test was used for the statistical analysis (*$P$ < 0.05).

C PC3-B2 cells were transfected with control or siCTH-1 and then were subjected to migration (16 h) and invasion (24 h) assays.

D Cell proliferation rates of PC3-B2 cells with CTH knockdown. Cell proliferation assay was performed by MTS reagent.

E Western blot analysis of the CTH levels in PC3 cells transfected with HA-CTH or pCMV-HA vector control.

F PC3 cells were transfected with HA-CTH or pCMV-HA vector control and then incubated for migration (16 h) and invasion assay (24 h).

G PC3 cells were pre-treated with 10 μM PAG for 16 h, and incubated for migration (16 h) and invasion assay (24 h) in the presence of 10 μM PAG.

Data information: (C, D, F, and G) Data shown represent the means ± SD ($n$ = 3 biological replicates). Student's $t$-test was used for the statistical analysis (*$P$ < 0.05; **$P$ < 0.01; ***$P$ < 0.001).

Source data are available online for this figure.

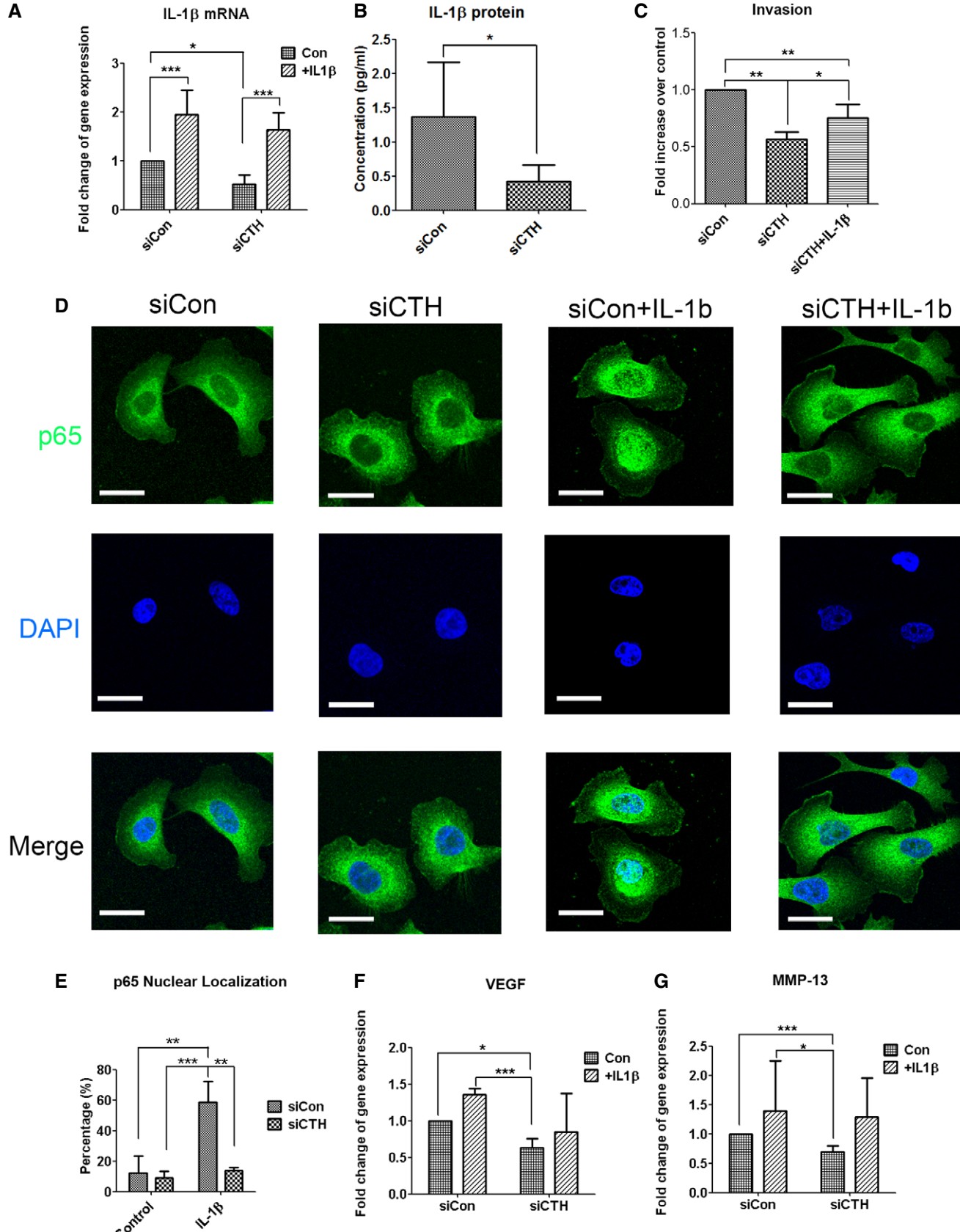

Figure 3.

**Figure 3.  CTH regulates cell invasion through IL-1β/NF-κB pathway.**

A       Real-time RT–PCR analysis for IL-1β mRNA levels in PC3 cells with control or CTH knockdown. Recombinant IL-1β (10 ng/ml) was added for 24 h in serum-free
        medium.
B       ELISA assay for IL-1β protein level in the culture medium of PC3 cells with control or CTH knockdown.
C       PC3 cells transfected with control or CTH knockdown were incubated for invasion (24 h) assay with recombinant IL-1β (10 ng/ml) added to the medium of the
        lower chamber.
D       PC3 cells transfected with control or CTH siRNA were exposed to IL-1β (20 ng/ml) for 1 h. Subcellular localization of p65 was detected by immunocytochemistry.
        Nuclei were counterstained with DAPI. The representative images are shown. Scale bars: 25 μm.
E       The percent of nuclear translocation for p65 was scored by counting the number of nuclear positive stained cells of p65 to the total number of cells in random
        microscopic fields.
F, G    Real-time RT–PCR analysis for VEGF (F) and MMP-13 (G) mRNA level in PC3 cells with control or CTH knockdown. Recombinant IL-1β (10 ng/ml) was added for
        24 h in serum-free medium.

Data information: All data shown (A–C and E–G) represent the means ± SD (*n* = 3 biological replicates). Student's *t*-test (B) or ANOVA followed by Tukey's post hoc test
(A, C, and E–G) was used for the statistical analysis. (*$P < 0.05$; **$P < 0.01$; ***$P < 0.001$).

suggesting the enzyme activity of CTH is required only for cell invasion. To further confirm whether CTH promoted cell migration is enzyme activity-independent, we generated CTH mutant with low enzymatic activity. Several polymorphic mutations on CTH are found in patients with cystathioninuria, and mutation on Q240E exhibits ~70-fold decrease in its kinetic activity [37]. Our data showed that the expression of this CTH$^{Q240E}$ mutant still induced cell migration in PC3 cells (Fig EV2F). However, the cell invasion was reduced in cells expressing this CTH$^{Q240E}$ mutant, as compared to the cells expressing CTH wild type (Fig EV2F). Due to the fact that enzymatic activity of CTH is not completely abolished in this Q240E mutant, there existed residual invasion promoting ability in the cells expressing this mutant. Overall, our data indicate that CTH protein promotes cell migration through an enzyme-independent pathway. In the future, we will try to identify proteins interacting with this CTH$^{Q240E}$ mutant to decipher the unknown mechanism of how CTH promotes cell migration.

### Knockdown of CTH suppressed cell invasion through inhibition of NF-κB-IL-1β-mediated signaling

The previous study demonstrated that cell survival prompted by NF-κB was mediated through sulfhydration of the p65 subunit [34]. We also observed increased endogenous p65 sulfhydration in PC3-B2 or PC3-B3 cells, as compared with PC3-T2 or PC3-T3 cells, while depletion of CTH by siRNA in PC3-B3 cells reduced the sulfhydration of p65 (Fig EV3A). To determine whether NF-κB sulfhydration is required for PC progression, we assessed the gene expression profile downstream from the NF-κB-mediated signaling. Among them, IL-1β, one of the pro-inflammatory cytokines, was significantly inhibited in PC3 cells with CTH knockdown (Fig 3A). IL-1β is known to play a positive autoregulatory loop through NF-κB-mediated signaling [38]. By adding human recombinant IL-1β to the culture medium, the reduced expression of IL-1β mRNA by CTH knockdown was restored (Fig 3A). Secreted IL-1β protein in the culture medium was reduced in PC3 cells with CTH knockdown (Fig 3B). More importantly, decreased cell invasion in PC3 cells with CTH knockdown was restored by treatment with recombinant IL-1β (Fig 3C), suggesting that the decreased cell invasion by CTH knockdown was likely due to blockage of the IL-1β production.

Nuclear localization of the p65 subunit is commonly used as a parameter of activation in the NF-κB-mediated pathway [39]. To assess whether CTH affects NF-κB transcriptional activities,

translocation of the p65 subunit was monitored by immunostaining. IL-1β stimulation caused nuclear translocation of the p65 subunit in almost sixty percent of the PC3 cell population, while the percent of cells with nuclear translocation decreased to a basal level in IL-1β stimulated cells with CTH knockdown (Fig 3D and E). The reduced nuclear translocation by IL-1β in CTH knockdown cells was further confirmed by subcellular fractionation of cytosol and nuclei (Fig EV3B and C). These observations suggest that CTH mediates NF-κB transcriptional activities through regulating p65 nuclear translocation. In addition, treatment of IL-1β did not change the expression of CTH (Fig EV3D), as well as p65 sulfhydration (Fig EV3E), suggesting that the sulfhydration of p65 is not stimulated by IL-1β-mediated signaling pathway.

Previous studies showed that both vascular endothelial growth factor (VEGF) and collagenase-3 (MMP-13) were upregulated by IL-1β stimulation [40,41]. We found that both VEGF mRNA expression and MMP-13 mRNA expression were significantly inhibited in PC3 cells with CTH knockdown, and treatment of IL-1β partially restored these reductions (Fig 3F and G), indicating the possible involvement of VEGF and MMP-13 in CTH-mediated signaling pathways.

### The H$_2$S-mediated sulfhydration of the NF-κB p65 subunit resulted in increased IL-1β production and enhanced cell invasion

To determine whether the CTH catalyzed product, H$_2$S, was involved in cell migration and invasion, sodium hydrosulfide (NaHS), or GYY4137, the donor of H$_2$S, was applied. In contrast to the rapid production of H$_2$S by NaHS, GYY4137 serves as a slow-releasing donor of H$_2$S [42]. Consistent with the previous observation, that only cell invasion requires the enzyme activity of CTH (Fig 2G), treatment with NaHS in a concentration range between 10 nM and 100 μM induced only cell invasion but not cell migration in PC3 cells (Fig 4A), and the same phenomenon was observed in 22Rv1 and C4-2 cells (Fig EV4A). Treatment with GYY4137 (1–10 μM) also significantly increased cell invasion in PC3 cells (Fig EV4B). The concentration of NaHS for maximal stimulation appeared to be between 10 nM and 10 μM. Dosages higher than 100 μM showed no effect on invasion (see Discussion). We further determined that reduced cell invasion by CTH knockdown in PC3 cells was abrogated in the presence of NaHS (Fig 4B), confirming that the CTH promotion of cell invasion was largely through H$_2$S production. We then measured sulfhydration of p65 by modified biotin switch assay. Our data indicated that sulfhydration of p65 (-SSH) was increased in the presence of NaHS, and this sulfhydration was depleted by

dithiothreitol (DTT) (Fig 4C), suggesting that the p65 modification by NaHS is S-S bond. Treatment of NaHS from 10 nM to 10 μM induced the highest levels of p65 sulfhydration in PC3 cells (Fig EV4C), implying that only small amount of H₂S, as low as 10 nM, is sufficient to induce sulfhydration of p65, resulting in the promotion of cell invasion (Fig 4A). Cells with positive nuclear staining of the p65 subunit increased to approximately forty percent in the presence of NaHS, as compared with the population of untreated cells (Fig EV4D and E). Moreover, serum-induced p65 nuclear translocation was reduced by CTH knockdown, and this reduction was restored in the presence of NaHS in PC3 cells (Appendix Fig S2). Treatment of NF-κB-specific inhibitors, SN50 and QNZ, attenuated NaHS and GYY4137-induced cell invasion (Figs 4D and EV4F), confirming our hypothesis that H₂S-mediated

cell invasion was due to NF-κB activation and its subsequent signaling pathways. The gene expression levels of IL-1β, MMP-13, and VEGF were substantially upregulated in the presence of NaHS (Fig 4E). Secreted IL-1β protein in the culture medium was also induced in PC3 cells treated with NaHS (10 nM–10 μM; Fig 4F). Meanwhile, consistent with previous the observation that H₂S mediated gene expression was due to coactivator ribosomal protein S3 (RPS3) [34], knockdown of RPS3 suppressed the expression levels of H₂S-induced genes, including IL-1β, MMP-13, and VEGF (Fig EV4G). Together, our data suggested that H₂S produced by CTH promoted cell invasion through IL-1β-NF-κB-mediated signaling pathways.

To determine whether sulfhydration of the NF-κB p65 subunit on cysteine 38 [34] is required for CTH/H₂S-mediated cell invasion, we

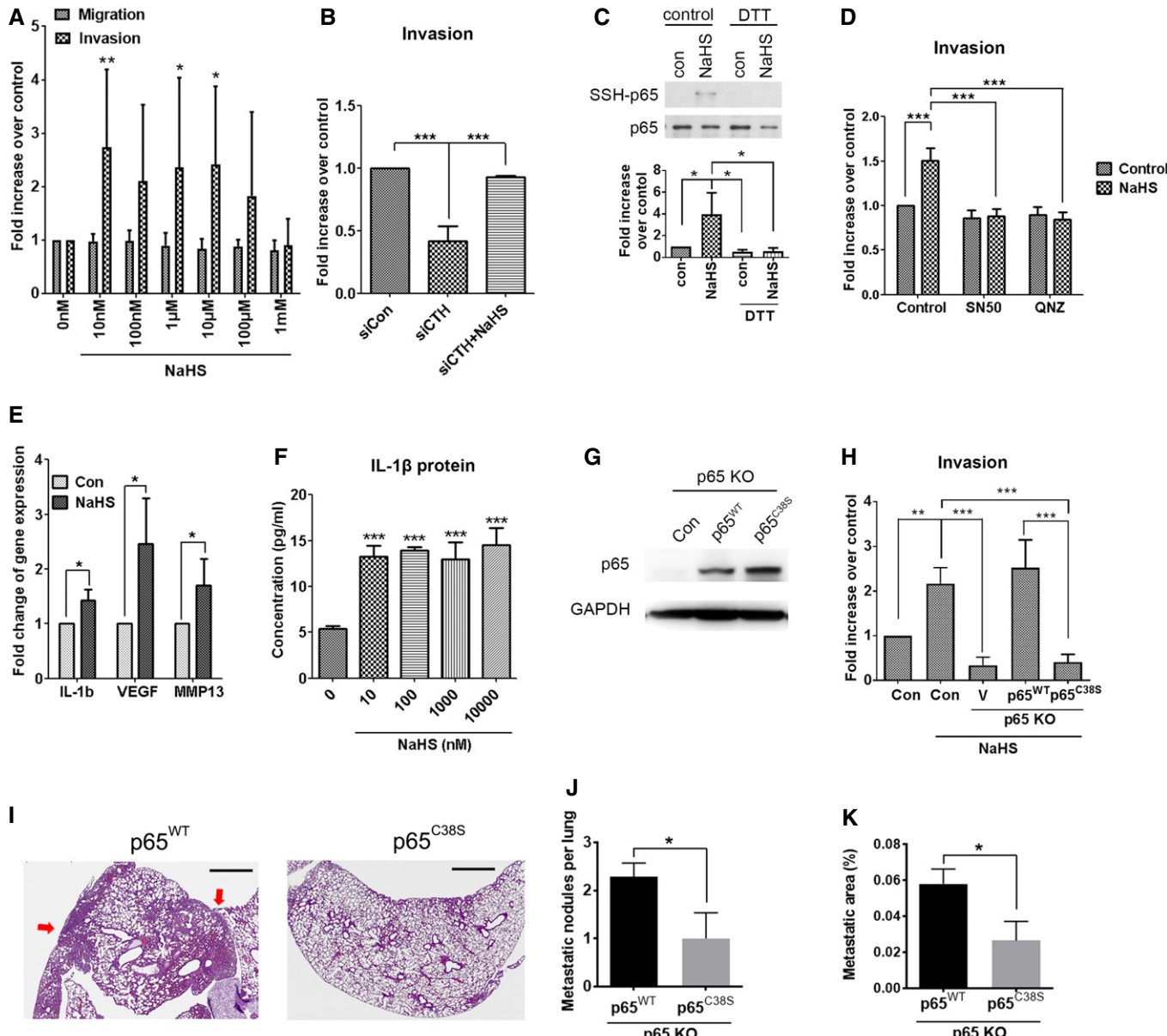

Figure 4.

◀

**Figure 4. H₂S promotes cell invasion through IL-1β/NF-κB pathway.**

A PC3 cells were incubated for migration (16 h) and invasion (24 h) assay. DMEM/10% FBS, together with various concentration of NaHS (10 nM–1 mM) acting as a chemoattractant.

B PC3 cells transfected with control or CTH siRNA were incubated for invasion (24 h) assay. DMEM/10% FBS, with or without NaHS (1 μM), acting as a chemoattractant.

C Top: PC3 cells treated with 100 μM NaHS for 30 min at 37°C. Cells were lyzed and treated with or without DTT for 10 min at 4°C before subjecting to the modified biotin switch assay with the antibody against p65 to detect S-sulfhydration. Down: Quantitative analysis of SSH-p65 protein level and normalized with total p65 level.

D PC3 cells were incubated for invasion assay for 24 h. DMEM/10% FBS, together with 1 μM NaHS and an NF-κB inhibitor (50 μg/ml SN50, or 100 nM QNZ), acting as a chemoattractant.

E Real-time RT–PCR analysis for IL-1β, VEGF, and MMP-13 mRNA levels in PC3 cells treated with 1 μM NaHS for 24 h.

F ELISA assay for IL-1β protein level in the culture medium of PC3 cells with various concentrations of NaHS (10 nM–10 μM) added for 24 h.

G Western blot analysis of the p65 expression in PC3 cells with p65 knockout and then transfected with p65 wild-type or C38S mutant.

H PC3 cells with p65 knockout with or without overexpression of p65 wild-type or C38S mutant were incubated for invasion assay for 24 h. DMEM/10% FBS was used as a chemoattractant.

I–K $1 \times 10^6$ PC3 cells with p65 knockout were transiently re-expressed p65 wild-type or C38S mutant and inoculated into the bloodstream via tail-vein injection. After 7 days of inoculation, mouse lung was collected to examine lung metastasis. (I) Images show H&E staining of the lungs from mice injected with p65 knockout PC3 cells re-expressing p65 wild-type or C38S mutant. Red arrows indicate metastatic nodules. Scale bar: 1 mm. (J, K) The total metastatic nodules per lung (J) and the percentage of the metastatic area per lung (K) were quantified. Data are presented as means ± SEM ($n = 7$ mice per group). Student's *t*-test was used for the statistical analysis (*$P < 0.05$).

Data information: (A–H) Data shown represent the means ± SD ($n = 3$ biological replicates). Student's *t*-test (E) or ANOVA followed by Tukey's post hoc test (A–D, F, and H) was used for the statistical analysis (*$P < 0.05$; **$P < 0.01$; ***$P < 0.001$).

Source data are available online for this figure.

established PC3 cell lines with CRISPR/Cas9-mediated knockout (KO) of p65 followed by re-expression of p65^wild-type or p65^C38S mutant by transient transfection (Fig 4G). The p65^wild-type or p65^C38S mutant was made with the in-frame mutation to replace sgRNA recognizing nucleotides, allowing these constructs to translate correctly and efficiently in the p65 KO PC3 cells. We then confirmed that sulfhydration of p65 was abolished in p65^C38S mutant in the presence of NaHS (Fig EV4H). Depletion of p65 attenuated NaHS-induced cell invasion (Fig 4H), indicating that H₂S-mediated cell invasion was due to NF-κB activation. Moreover, re-expression of p65^wild-type, but not p65^C38S mutant, restored invasion abilities (Fig 4H), suggesting the sulfhydration site on cysteine 38 was required for p65-mediated cell invasion. Moreover, nuclear translocation of p65 was observed in the p65 KO PC3 cells with re-expression of the p65^wild-type, but not the p65^C38S mutant, in the presence of IL-1β (Fig EV4I and J). Our data indicated that sulfhydration on the cysteine 38 of the NF-κB p65 subunit was required for its nuclear translocation and subsequent cell invasion ability. To examine whether p65 sulfhydration on cysteine 38 was involved in the progression of prostate cancer metastasis, we transiently transfected p65^wild-type or p65^C38S mutant in p65 KO PC3 cells, and then, those cells were inoculated into the mice via tail-vein injection. The p65 KO PC3 cells with the p65^C38S mutant re-expression showed decreased number of lung-metastatic nodules (Fig 4I and J) and reduced percentage of lung-metastatic area (Fig 4I and K), as compared to the p65^wild-type re-expression group, suggesting that p65 sulfhydration could be directly involved in the promotion of cancer metastasis.

**Knockdown of CTH suppressed tumor growth and reduced the incidence of paraaortic lymph nodes and bone metastases in the mouse orthotopic implantation model**

To investigate the role of CTH during PC progression, orthotopic implanted xenografts in nude mice were utilized. PC3 cells stably expressing shCTH or shCon (Fig 5A) were orthotopically injected into

the prostate capsules of nude mice. Mice were dissected 60 days after injection. In the CTH knockdown group, both tumor weight and tumor size decreased substantially to about 50% of those found in the shCon group (Fig 5B and C; Appendix Fig S3A). Meanwhile, the mouse weight slightly dropped in the shCon group (Appendix Fig S3B). We then checked the incidence of paraaortic lymph node metastases through histological evaluation. Metastases in paraaortic lymph nodes were observed in 80% of mice in the shCon group compared with only 40% in the shCTH group (Fig 5D). Most of the paraaortic lymph nodes were enlarged (Appendix Fig S3C) and located inside the lymph node cavity in the shCon mice (Appendix Fig S3D; left panel), while most of the shCTH mice displayed smaller sized lymph nodes (Appendix Fig S3C) with no metastases found inside the lymph nodes (Appendix Fig S3D; right panel).

After we cultured the cells derived from bone marrow of the mice with orthotopic implantation, PC3 cells were detectable in 2 of 10 mice in the shCon group and undetectable in the shCTH group (Fig 5E). Those bone-derived PC3 cells were resistant to puromycin, confirming their lineage from the original injected PC3 cells that stably expressed puromycin-resistant plasmid. As expected, the expression levels of CTH in the orthotopically implanted tumors were reduced in the shCTH groups, as compared with the shCon group (Fig 5F). In summary, our results indicated that the reduction of CTH expression in PC3 cells decreased tumor growth, as well as paraaortic lymph node and bone metastasis.

To investigate whether CTH is required for PC progression, we stably expressed CTH in DU145 cells. DU145 line was originally derived from brain, and not bone, metastasis [43], and the expression level of CTH was much lower in DU145 cells, as compared to other prostate cancer lines (Fig EV1C). The control and CTH stable overexpressing lines were established (Fig EV5A) and orthotopically injected into the prostate capsules of nude mice. In the CTH overexpressing group, the mouse weight was slightly decreased as compared to the control group (Fig EV5B), and tumor size was significantly increased in comparison with the control group (Fig EV5C and E). The incidence of paraaortic lymph node

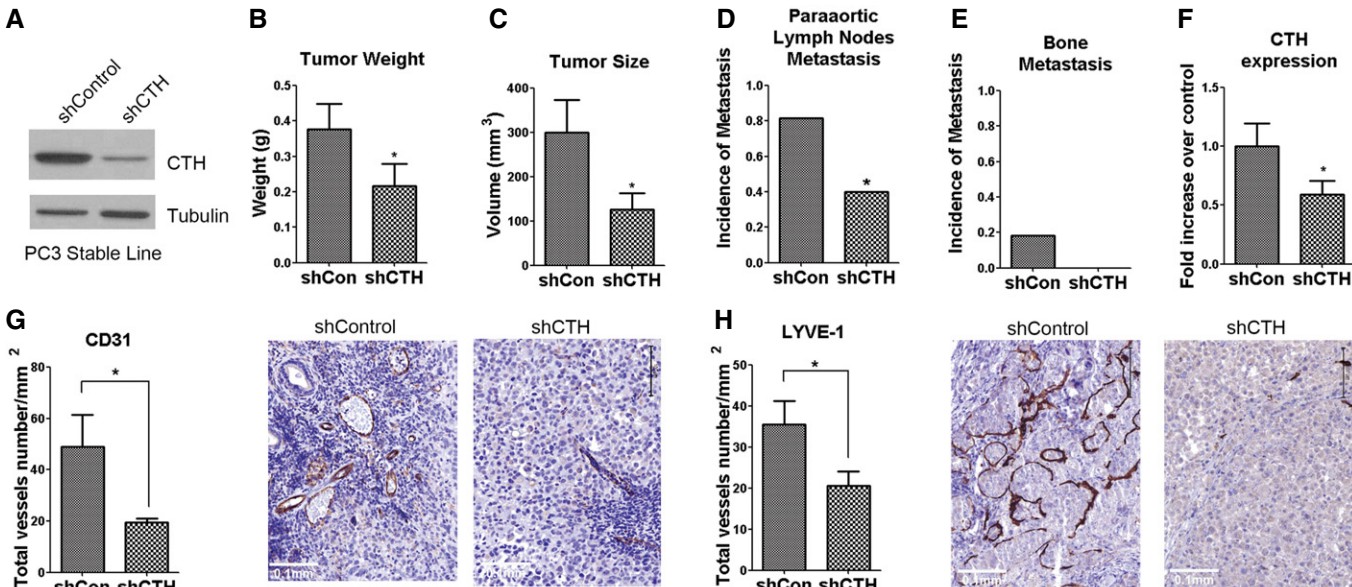

**Figure 5.  Knockdown of CTH suppresses tumor growth and distant metastasis in the orthotopic implantation mouse model.**

A    Western blot analysis of the CTH expression in PC3 cells stably expressing control or CTH shRNA.

B–E    $1 \times 10^6$ PC3 cells with control or CTH knockdown were orthotopically injected into the mouse prostate for 60 days. The tumor weight (B) and tumor volume (C) were compared. Incidences of paraaortic lymph nodes metastasis (D) and bone metastasis (E) are shown. Data are presented as means ± SEM ($n$ = 10 mice per group).

F    Real-time RT–PCR analysis for CTH mRNA levels in PC3 orthotopic tumors from shCon or shCTH group. Data shown represent normalized means ± SEM ($n$ = 10 mice per group).

G, H    Immunohistochemical analysis of paraffin-embedded xenograft prostate tumor using CD31 (G) and LYVE-1 (H) antibody. The total vessel number per $mm^2$ was quantified. Scale bars: 0.1 mm. Data are presented as means ± SEM ($n$ = 4–5 mice per group).

Data information: (B–H) Student's $t$-test was used for the statistical analysis (*$P < 0.05$).
Source data are available online for this figure.

metastasis was significantly increased in the CTH group as compared to the control group (Fig EV5D, F and G). However, bone metastasis was not detected in either group under our experimental duration. Overall, our data suggest that CTH is capable of promoting tumor growth, as well as lymph node and bone metastasis in certain PC lines such as PC3. However, CTH may be required, but not sufficient, for promoting bone metastasis.

$H_2S$ is an endogenous stimulator of angiogenesis by accelerating endothelial cell growth, cell migration, and capillary morphogenesis [44]. To investigate whether CTH knockdown affects angiogenesis and lymphangiogenesis, we examined the expression of corresponding endothelial cell markers using respective antibodies, anti-cluster of differentiation 31 (CD31), and anti-lymphatic vessel endothelial hyaluronan receptor 1 (LYVE-1), in primary orthotopic tumors with immunohistochemical analysis. Both CD31- and LYVE-1-positive vessel structures were present in the orthotopic tumors (Fig 5G and H). In tumors with CTH knockdown, the numbers of vessel-like structures revealed by CD31 or LYVE-1 antibody staining were significantly reduced, as compared with the shCon group (Fig 5G and H). Our data suggested that CTH/$H_2S$ signaling was important for tumor angiogenesis and lymphangiogenesis. Consistent with this *in vivo* observation, HUVEC cells cultured with the conditional medium derived from PC3-B2 cells with CTH knockdown also showed a considerably lower percentage of tube formation *in vitro* (Appendix Fig S4).

## Discussion

In the present study, we identified a signaling cascade mediated by CTH/$H_2S$ to promote PC progression and metastasis (Fig 6). Increased expression of CTH in bone-metastatic PC cells induced a change in $H_2S$ level, resulting in the activation of IL-1β/NF-κB-mediated signaling to promote cell invasion, angiogenesis, lymphangiogenesis, tumor growth, and metastasis. Our study implies that $H_2S$ and its generating enzyme, CTH, may serve as potential therapeutic targets for PC metastasis intervention.

Previous studies presented controversial results about $H_2S$ in cancer progression [16]. Increased endogenous $H_2S$ in the malignant cells enhanced tumor cell proliferation, drug resistance, and angiogenesis [18,45], while high doses of exogenous $H_2S$ treatment weakened tumors by suppressing tumor cell growth [46]. Literature studies described the physiological concentrations of $H_2S$ within a wide range between 10 nM and 300 μM [47]. Here, our data indicated that $H_2S$ could promote cell invasion ability in a concentration range from 10 nM to 100 μM, and higher doses of $H_2S$ showed no effects on cell invasion, as compared with the control (Fig 4A). Consistent with the previous observation that endogenous $H_2S$ played a role in promoting oncogenesis, our data indicated that $H_2S$ enhanced cell invasion only at the physiological concentration range.

In this study, we showed that CTH expression promoted both cell migration and invasion (Fig 2C and F). However, treatment with

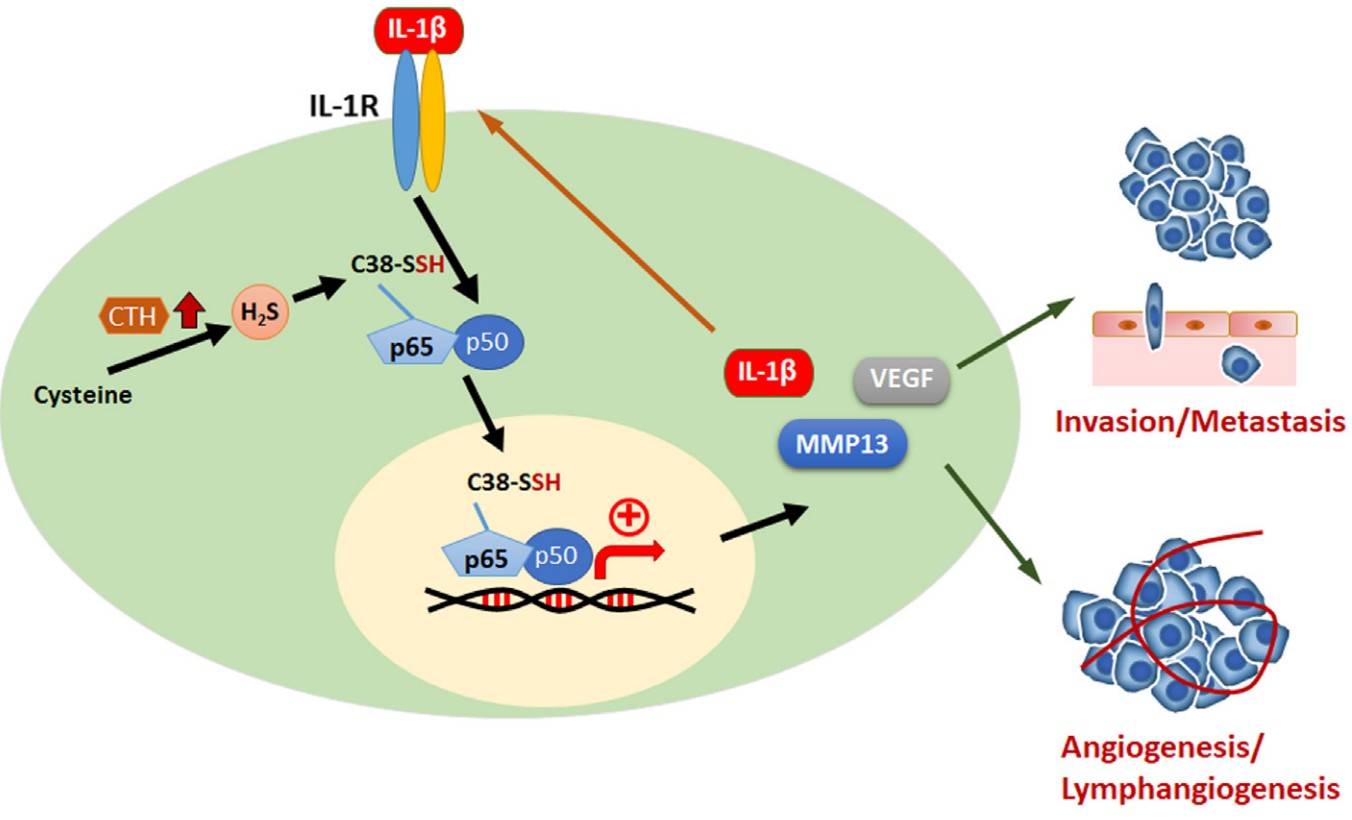

**Figure 6. Current working model of CTH/H₂S-mediated signaling in PC progression and distant metastasis.**

H₂S enhanced only cell invasion but not cell migration (Fig 4A). Our data also indicated that treatment with CTH-specific enzymatic inhibitor, PAG, suppressed only cell invasion (Fig 2G). In contrast, the expression of CTH$^{Q240E}$, the mutant form of CTH with lower enzymatic activity [37], induced only cell migration, but not cell invasion (Fig EV2F), suggesting that the enzyme activity of CTH promoted cell invasion mainly through its derivative product, H₂S, mediated signaling pathways. Conversely, CTH-induced cell migration was regulated through an enzyme-independent pathway. Additional studies are required to unveil the underlying mechanism of how CTH modulates cell migration.

NF-κB activation requires translocation of NF-κB subunits, p65 and p50, from the cytosol to the nucleus [48,49]. Nuclear translocation of the NF-κB is initiated by the signal-induced degradation of IκB proteins through activation IκB kinase (IKK). The degradation of IkB thus releases NF-κB to translocate into the nucleus and activate gene transcriptions [50]. Here, our data showed that blocking p65 sulfhydration resulted in the attenuation of p65 nuclear translocation induced by IL-1β (Figs 3D and EV4I), suggesting sulfhydration of p65 might be involved in the nuclear import of the p65 subunit. We also noticed that treatment with H₂S alone only induced modest nuclear translocation of p65 (Fig EV4D), and this induction is incomparable to the level of IL-1β-induced nuclear translocation of p65 (Fig 3D). Based on these observations, we believe that p65 sulfhydration by H₂S is not enough to stimulate the p65 nuclear translocation since NF-κB complex may still interact with the inhibitory protein IkB. Additional signals, such as IL-1β, are required to activate IKK through

phosphorylation, resulting in the degradation of IkB to release p65. The p65 sulfhydration may be required for the interaction between p65 and nuclear transport proteins to facilitate nuclear import. More research is needed to determine the exact role of p65 sulfhydration in regulating NF-κB activity.

Although H₂S is an endogenous stimulator of angiogenesis [44], the underlying mechanism remains unclear. Here, we demonstrated that treatment with H₂S induced the expression of IL-1β (Fig 4E and F). IL-1β is a known pro-angiogenic cytokine during cancer progression through induction of VEGF [51]. Coincidentally, our data also indicated that H₂S induced VEGF and MMP-13 expression (Fig 4E). Taken together, H₂S likely stimulates angiogenesis through IL-1β-VEGF signaling pathway.

Clinical studies revealed that lymph node metastases usually reflect poor prognosis for PC patients [52], and the metastatic routes for PC were often first through lymphatic vessels to the regional lymph nodes. Tumor-associated lymphangiogenesis, therefore, is crucial to facilitate lymph node metastasis [53]. Here, we observed that lymphangiogenesis was significantly reduced in CTH knockdown tumors (Fig 5H). Simultaneously, we observed lower incidence of paraaortic lymph node metastases in mice bearing tumors with CTH knockdown (Fig 5D), implying that reduced lymphangiogenesis by CTH knockdown may result in reduced lymph node metastasis. Overall, our data suggested a novel mechanism whereby CTH/H₂S may facilitate PC metastasis through enhancing lymphangiogenesis in the tumor microenvironment.

# Materials and Methods

### Cell culture and transfections

The RWPE-1 and C4-2 cell lines were kindly provided by Dr. Hsing-Jien Kung. The PC3, LNCaP, 22Rv1, and DU145 cell lines were obtained from the Bioresource Collection and Research Center (BCRC), Taiwan. RWPE-1 was maintained in Keratinocyte Serum-Free Medium (K-SFM; Invitrogen) with 0.05 mg/ml bovine pituitary extract (BPE) and 5 ng/ml human recombinant epidermal growth factor (EGF) at 37°C in 5% $CO_2$. LNCaP, C4-2, 22Rv1, and DU145 were maintained in RPMI 1640 medium (RPMI; Invitrogen) with 10% FBS at 37°C in 5% $CO_2$. PC3 cells were maintained in Dulbecco modified Eagle medium (DMEM; Invitrogen) with 10% FBS at 37°C in 5% $CO_2$. Plasmids were transfected using TransIT 2020 or TransIT X2 (Mirus Bio), while short interfering RNA (siRNA) was transfected using Lipofectamine RNAiMAX (Invitrogen) or TransIT X2 (Mirus Bio). PC3 cell lines stably expressing control or CTH shRNA were selected using puromycin (Sigma-Aldrich) and established as mass culture. DU145 cell lines stably expressing PCDNA or PCDNA-CTH were selected using G418 (Sigma-Aldrich) and established as mass culture.

### Cell knockout and mutagenesis

To generate p65 knockout cells, PC3 cells were co-transfected with pU6-PGK-puro-sgRELA (Sigma-Aldrich) and p5W-Cas9.pBsd plasmids (National RNAi Core Facility, Academia Sinica, Taiwan) by a TransIT-X2 reagent (Mirus). Knockout PC3 cells were selected by using 4 μg/ml blasticidin and 3 μg/ml puromycin and established as mass culture. The p65 subunit of NF-κB in-frame mutation (GA<u>TATC</u>GAGGTGTATTTCACGG) to replace sgRNA recognizing nucleotides (GA<u>CATT</u>GAGGTGTATTTCACGG) and NF-κB in-frame mutation plus C38S mutation plasmid was generated by site-directed mutagenesis using PfuUltra II Fusion HS DNA polymerase (Agilent Technologies). PCDNA-p65 was the template plasmid. CTH-Q240E mutation was generated by site-directed mutagenesis from the pCMV-HA-CTH plasmid. All primer sequences used in this study are provided in Appendix Table S1.

### Nucleotides and reagents

The CTH-specific and RPS-specific siRNAs were purchased from MDBio, Inc (Qingdao, China). The pRFP-C-RS-shCTH construct was purchased from OriGene (Rockville, MD). Detailed siRNA and shRNA sequences are listed in Appendix Table S1. The full-length CTH was amplified by RT–PCR of cDNAs from PC3 cells and cloned into pCMV-HA and PCDNA3.1 vectors. The full-length p65 was amplified by RT–PCR of cDNAs from PC3 cells and cloned into PCDNA3.1. All primer sequences used in this study are provided in Appendix Table S1. Antibodies against CTH were purchased from Abnova (Taiwan) or Santa Cruz Biotechnology. Antibodies against CBS, α-tubulin, GAPDH, and actin were purchased from Santa Cruz Biotechnology. Antibodies against NF-κB p65 subunit were purchased from Cell Signaling. Anti-3MST was purchased from GeneTex (Taiwan). Anti-Lamin B2 was purchased from Millipore. The $H_2S$ donors, NaHS (for fast release) or GYY4137 (for slow release), were purchased from Sigma-Aldrich. The CTH inhibitor, DL-propargylglycine (PAG), was obtained from Cayman Chemical. Human recombinant IL-1β was obtained from BioLegend. NF-κB inhibitor, SN50, was obtained from Merck Millipore, and QNZ was obtained from Enzo Life Sciences.

### Quantitative real-time PCR

RNA was extracted from cells using TRIzol (Invitrogen) following protocols supplied by the manufacturer. First-strand cDNA was generated by ReverTra Ace (Toyobo, Japan) using oligo-dT. Real-time RT–PCR was performed on a CFX96 real-time PCR detection system (Bio-Rad). The KAPA SYBR FAST Universal qPCR Kit (KAPA Biosystems, MA) was used. The mRNA levels were normalized to that of actin. All primer sequences used in this study are provided in Appendix Table S1. Experiments were repeated at least three times.

### $H_2S$ measurements (lead sulfide method)

For direct measurement of $H_2S$, we leveraged the specific reaction between $H_2S$ and lead acetate to form a black precipitate (lead sulfide) that could be trapped and visualized on filter paper containing lead acetate [54]. Briefly, 100–300 μg freeze–thaw homogenate of cells was lysed in a passive lysis buffer (Promega), supplemented with 10 mM Cys and 10 μM PLP. A lead acetate detection paper (Sigma) was placed above the liquid phase in a closed tube and incubated for 2–5 h at 37°C until lead sulfide darkened the paper. The intensity of the lead sulfide deposit was quantified by ImageJ. Data represented as fold changes relative to the control. All experiments were repeated three times. Histograms were the means ± SD from the three independent experiments.

### Modified biotin switch (S-sulfhydration) assay

The assay was performed in accordance with a published paper [5], with modifications. Briefly, PC3 cells were treated with 1 nM–1 mM NaHS for 30 min at 37°C in culture medium, and then homogenized in a HEN buffer [250 mM Hepes-NaOH (pH 7.7), 1 mM EDTA, and 0.1 mM neocuproine], supplemented with 100 μM deferoxamine, and centrifuged at 13,000 *g* for 30 min at 4°C. Cell lysates (1 mg) were treated with or without 100 μM DTT for 10 min at 4°C. The lysates were then added to a blocking buffer (HEN buffer adjusted to 2.5% SDS and 20 mM MMTS) at 50°C for 20 min with frequent vortexing. The proteins were precipitated by 100% acetone at −20°C for 20 min. After acetone removal, the proteins were resuspended in a HENS buffer (HEN buffer adjusted to 1% SDS), and then, 4 mM biotin-HPDP (EZ-link, ThermoFisher) was added. After incubation for 3 h at RT, biotinylated proteins were precipitated by streptavidin agarose beads at 4°C overnight and then washed with the HENS buffer. The biotinylated proteins were analyzed by SDS–polyacrylamide gel electrophoresis and subjected to Western blot analysis with antibodies against NF-κB p65 subunit (Cell Signaling). All experiments were repeated three times. Histograms were the means ± SD from the three independent experiments.

### Nuclear/cytosol fractionation

Cells were washed with ice-cold PBS twice and harvested in PBS. One-fourth of the cells were transferred to a new Eppendorf. PBS

was removed after centrifuging the cells at 1,500 *g* for 3 min. The total lysate was collected by adding a RIPA lysis buffer (10 mM Tris, pH 7; 150 mM NaCl; 1% Triton X-100; 1% Na deoxycholate; 1% SDS; 1× protease inhibitor) into one-fourth of the cells. The remaining cells were lysed by an NP-40 lysis buffer (1× TBS; 0.5% NP-40; 1× protease inhibitor). The suspension, containing cytosolic proteins, was transferred into a new Eppendorf. The remaining pellets were washed with an NP-40 lysis buffer twice and then lysed by a RIPA lysis buffer as a nuclear fraction. Alpha-tubulin antibody (Santa Cruz Biotechnology) was used as a cytosolic marker, while antibody against Lamin B2 (Millipore) was used as a nuclear marker. All experiments were repeated three times. Histograms were the means ± SD from the three independent experiments.

### Western blot analysis

Cells were lysed in a 10 mM Tris buffer, pH 7.4, containing 0.15 M NaCl, 1% Triton X-100, 1 mM EDTA, and protease inhibitor mixture (Roche). The cell lysates were resolved in an 8–12.5% SDS-polyacrylamide gel, transferred onto PVDF membrane and probed with antibodies. All experiments were repeated three times.

### Transwell cell migration and cell invasion assay

Cell migration and invasion Boyden chamber assays were performed as previously described [55,56]. Cell migration was assayed in 8.0-mm Falcon Cell Culture Inserts (Corning), and for the cell invasion assay, the BD BioCoat Matrigel Invasion Chamber was applied (Corning). Briefly, $1 \times 10^5$ or $5 \times 10^4$ cells were suspended in DMEM (300 μl) and placed in an upper transwell of 0.3 cm$^2$ in area. The bottom well was filled with 500 μl DMEM with 10% FBS. After incubation for 16–24 h (Migration) or 24 h (Invasion), cells on the upper side of the inserts were removed by cotton swabs, and cells on the underside were fixed and stained with crystal violet. Photos of three regions were taken from duplicated inserts, and the numbers of cells were counted using ImageJ (NIH, US). All experiments were repeated three times. Data were the means ± SD from the three independent experiments.

### Cell proliferation assay

The cell proliferation assay was measured with CellTiter 96 AQueous One Solution (Promega). The assay was performed according to the methods described in the manufacturer manual. Briefly, cells ($1 \times 10^3$ cells) were seeded in 96-well plates and incubated for various times, at defined time points, and 20 μl of CellTiter 96 AQueous One Solution Reagent was added and incubated for 2 h at 37°C. The quantity of formazan product, which is directly proportional to the number of living cells in the culture, was measured by absorbance at 490 nm with a 96-well plate reader. All experiments were repeated three times. Data were the means ± SD from the three independent experiments.

### GSH/GSSG assay

Cells were seeded $5 \times 10^3$ at 96-well overnight then lysed by a glutathione/oxidized glutathione lysis buffer for 30 min at RT. 50 μl cellular lysis was taken for luciferin reaction and detection, which

followed the GSH/GSSG-Glo Assay Kit (Promega) standard protocol. All experiments were repeated three times. Data were the means ± SD from the three independent experiments.

### Immunofluorescence

Cells were seeded $1 \times 10^4$ on fibronectin-coated cover-slides overnight and then cultured at serum-free (SF) medium for 4 h. For cell signaling activation, 20 ng/ml IL-1β, 100 μM NaHS, or 10%FBS in SF-medium was used for 30-min or 1-h treatment. Those cover-slides of cells were washed with PBS, fixed at 4% paraformaldehyde for 30 min and 0.5% Triton X-100 for 5 min at RT, and then, 20% FBS in PBS solution for 1 h at RT was used for blocking. After fixed and blocking steps, cells were incubated p65 antibody (Cell Signaling; 1:100 dilution) in 1% BSA-PBS solution at 4°C overnight. The next day, cover-slides were washed with PBS for three times and stained with Alexa Fluor 488-coupled antibody (Invitrogen; 1:500 dilution) or DAPI for 1 h at RT; then, cover-slides were sealed for microscopic observation. Subcellular localization of p65 was detected by immunofluorescence (IF) using antibodies against p65 nuclei were counterstained with DAPI. Images were acquired using confocal microscopy (TCS SP5, Leica). The percentage of cells with positive p65 nuclei staining were counted. At least 25 (Fig EV4J) or 50 cells (all the other experiments) per condition were included in each experiment. To avoid bias, all cells imaged from a single field of view were analyzed. All experiments were repeated three times. Data were the means ± SD from the three independent experiments.

### Endothelial cell tube formation assay

HUVEC cells, which spontaneously form capillary tubes when seeded on a Matrigel basement membrane matrix, were used to assay the effects of different conditioned mediums on angiogenesis. To coat the plates with the basement membrane, 50 μl of Matrigel was loaded on each well of a 96-well dish. $1.5 \times 10^4$ HUVEC cells were suspended in cancer cell-derived conditional mediums or fresh mediums, and then seeded on the solidified Matrigel in a well. After a 16-h incubation, the network of the tube formed by the cells was photographed for further analysis. The number of tube-like structures with closed networks of vessel-like tubes was counted. The quantification of the tube network was based on the number of nodes formed by at least three segments. All experiments were repeated three times. Data were the means ± SD from the three independent experiments.

### IL-1β detection

Culture supernatants from the serum-free medium of $1 \times 10^6$ PC3 cells incubated for 24 h were harvested. The level of IL-1β released was measured by an ELISA kit in accordance with the manufacturer instructions (eBioscience). All experiments were repeated three times. Data were the means ± SD from the three independent experiments.

### Mouse orthotopic implantation and tail-vein injection model

Male athymic BALB/c nude mice were purchased from NLAC, Taiwan, and Bio LASCO Taiwan Co. Eight- to twelve-week-old male

mice were anesthetized with isoflurane and placed in the supine position. $1 \times 10^6$ PC3-T2, B2, T3, and B3 cells, PC3 cells stably expressing CTH or control shRNA, or DU145 cells stably expressing PCDNA or PCDNA-CTH, in 15ul were mixed with Matrigel in a 1:1 ratio and then injected into the prostate as described [35,57]. The mice were examined 15–60 days after tumor cell injection. Original tumors, paraaortic lymph nodes (LN), and bone marrow (BM) were harvested for hematoxylin and eosin (H&E) staining or *in vitro* BM culture. For the tail-vein injection experiment, $1 \times 10^6$ PC3 cells with p65 knockout were transiently re-expressed p65$^{wild-type}$ or p65$^{C38S}$ and inoculated into the bloodstream via tail-vein injection. Lung tissues were collected after 1 week of injection for H&E staining. Tissue specimens (LN, prostate tumor, and lung) were fixed, paraffin-embedded, serially sectioned, and stained with H&E by a pathology core lab (NHRI, Taiwan). The BM cells from mouse thigh bones were collected and cultured in RPMI or DMEM with 10% FBS to detect bone-metastatic cells from the primary tumor in the prostate. Data of all *in vivo* experiments were combined from two independent experiments. All procedures were done according to the protocol by approved Institutional Animal Care and Use Committee of National Health and Research Institutes, Taiwan (NHRI-IACUC-100130-A; NHRI-IACUC-108015-A).

**Immunohistochemistry**

The paraffin-embedded tissue sections of human PC specimens, including primary tumors, were obtained from commercial PC tissue array slides (US Biomax, Inc). The slides were sent to a pathology core lab (NHRI, Taiwan) for staining with CTH antibody (Abnova) by using an automatic slide stainer from BenchMark XT (Ventana Medical Systems, AZ).

**Retrieving RNA-seq data from The Cancer Genome Atlas**

The normalized RNA-seq data containing 469 prostate cancer (PC) cases, 50 normal cases, 174 pancreatic adenocarcinomas (PAAD), 528 lower grade glioma (LGG) cases, and patients' survival status were retrieved from The Cancer Genome Atlas (TCGA) (https://cancergenome.nih.gov). The clinical data of PC are summarized in Appendix Table S2. A total of 469 PC, 174 PAAD, and 528 LGG cases were included in the Kaplan–Meier survival analysis in this study.

**Statistical analyses**

The statistical analyses were all assessed by GraphPad Prism using Student's *t*-test to compare two means. ANOVA followed by Tukey's post hoc test was used for the statistical analysis when more than two means were compared (*$P < 0.05$; **$P < 0.01$; ***$P < 0.001$).

**Expanded View** for this article is available online.

## Acknowledgements

We thank the Pathology Core Lab of the National Health Research Institutes for all of the H&E and IHC staining. Yi-Hsiang carried out his thesis research under the auspices of the Graduate Program of Biotechnology in Medicine, National Tsing Hua University and National Health Research Institutes. This work is financially supported by the "Chinese Medicine Research Center, China Medical University" from The Featured Areas Research Center Program within the framework of the Higher Education Sprout Project by the Ministry of Education(MOE) in Taiwan (CMRC-CHM-6), National Health Research Institutes (NHRI) (NHRI 06A1-MGPP09-014), Ministry of Health and Welfare, (MOHW103-TDU-M-221-123017), and Ministry of Science and Technology (MOST), Taiwan (MOST 104-2320-B-039-055-MY3, MOST 104-2320-B-039-054-MY3, MOST 106-2811-B-039-004, MOST 106-2314-B-007-007, MOST 107-2320-B-007-003-MY3, MOST 108-2314-B-007-003-MY3).

## Author contributions

J-TH, Y-HW, W-LC, R-HW, M-CK, Y-RP, and K-TL carried out the experimental work. S-HC and K-WT assisted in the TCGA clinical sample analysis. H-JK provided reagents, ideas, and suggestions. K-TL designed experiments and wrote the manuscript. K-TL and L-HW obtained funding for this project, directed and supervised the research, as well as revised and approved the manuscript. K-TL and L-HW are co-corresponding authors.

## Conflict of interest

The authors declare that they have no conflict of interest.

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
