## [Review Process File · EMBO Reports]

Dysregulation of cystathionine γ -lyase promotes prostate cancer progression and metastasis

Yi-Hsiang Wang, Jo-Ting Huang, Wen-Ling Chen, Rong-Hsuan Wang, Ming-Chien Kao, Yan-Ru Pan, Shih-Hsuan Chan, Kuo-Wang Tsai, Hsing-Jien Kung, Kai-Ti Lin, and Lu-Hai Wang

Review timeline:

Submission date:	7 December 2018
Editorial Decision:	16 January 2019
Revision received:	17 July 2019
Editorial Decision:	5 August 2019
Revision received:	6 August 2019
Accepted:	8 August 2019

Editor: Achim Breiling

Transaction Report:

1st Editorial Decision

16 January 2019

Thank you for the submission of your research manuscript to EMBO reports. We have now received reports from the three referees that were asked to evaluate your study, which can be found at the end of this email.

As you will see, all referees think the manuscript is of interest, but requires revisions to allow publication in EMBO reports. All three referees have a number of concerns and/or suggestions to improve the manuscript, which we require all to be addressed in a revised manuscript. As the reports are below, I will not detail them here. However, we think that it will be of particular importance that the comments of referee #1 are addressed experimentally, including the use of another human PCa cell line (comment 4 of referee #1). Further, please have your revised manuscript carefully proof-read by a native speaker.

Given the constructive referee comments, I would like to invite you to revise your manuscript with the understanding that all referee concerns must be addressed in the revised manuscript and/or in a detailed point-by-point response. Acceptance of your manuscript will depend on a positive outcome of a second round of review. It is EMBO reports policy to allow a single round of revision only and acceptance or rejection of the manuscript will therefore depend on the completeness of your responses included in the next, final version of the manuscript.

Revised manuscripts should be submitted within three months of a request for revision; they will otherwise be treated as new submissions. Please contact us if a 3-months time frame is not sufficient for the revisions so that we can discuss the revisions further.

Supplementary/additional data: The Expanded View format, which will be displayed in the main HTML of the paper in a collapsible format, has replaced the Supplementary information. You can submit up to 5 images as Expanded View. Please follow the nomenclature Figure EV1, Figure EV2

etc. The figure legend for these should be included in the main manuscript document file in a section called Expanded View Figure Legends after the main Figure Legends section. Additional Supplementary material should be supplied as a single pdf labeled Appendix. The Appendix includes a table of content on the first page, all figures and their legends. Please follow the nomenclature Appendix Figure Sx throughout the text and also label the figures according to this nomenclature.

For more details please refer to our guide to authors:
<http://embor.embopress.org/authorguide#manuscriptpreparation>

Important: All materials and methods should be included in the main manuscript file.

See also our guide for figure preparation:
http://www.embopress.org/sites/default/files/EMBOPress_Figure_Guidelines_061115.pdf

Please also format the references according to EMBO reports style. See:
<http://embor.embopress.org/authorguide#referencesformat>

Regarding data quantification and statistics, can you please specify, where applicable, the number "n" for how many independent experiments (biological replicates) were performed, the bars and error bars (e.g. SEM, SD) and the test used to calculate p-values in the respective figure legends. Please provide statistical testing where applicable. See:
<http://embor.embopress.org/authorguide#statisticalanalysis>

Please add scale bars to all microscopic images, without any text on or near them, and of comparable thickness. Please define the scale bars in the corresponding figure legend.

As the Western blots shown in the manuscript are significantly cropped, we ask you to submit the original source data, also with the aim of making primary data more accessible and transparent to the reader. The source data will be published in a separate source data file online along with the accepted manuscript and will be linked to the relevant figure. Please submit the source data (scans of entire gels or blots) of the Western blot panels (main figures, EV figures and Appendix figures) together with the revised manuscript. Please include size markers for scans of entire gels, label the scans with figure and panel number, and send one PDF file per figure.

- a complete author checklist, which you can download from our author guidelines (<http://embor.embopress.org/authorguide#revision>). Please insert page numbers in the checklist to indicate where the requested information can be found.
- a letter detailing your responses to the referee comments in Word format (.doc)
- a Microsoft Word file (.doc) of the revised manuscript text
- editable TIFF or EPS-formatted single figure files in high resolution (for main figures and EV figures)

Please also note that we now mandate that all corresponding authors list an ORCID digital identifier that is linked to his/her EMBO reports account!

I look forward to seeing a revised version of your manuscript when it is ready. Please let me know if you have questions or comments regarding the revision.

REFeree REPORTS

Referee #1:

Here the authors set out to investigate the role of cystathionine- γ -lyase (CTH/CSE), an enzyme that generates H₂S from cysteine, in NF- κ B signaling and the formation of bone metastases in prostate cancer (PCa). They started by comparing RNA microarray expression patterns of human PC3 PCa

cells grown as an orthotopic tumor in immunocompromised mice with PC3 cells that were isolated from bone marrow after their dissemination from a primary tumor, analyzing a total of seven PC3-T (primary) lines with paired PC3-B (bone) lines. They noted that CTH RNA and protein were increased in the PC3-B cells in 5/7 pairs, together with elevated levels of H₂S, and that PC3-B cells had greater migratory and invasive properties. siRNA-mediated knockdown of CTH reduced H₂S production and both invasion and migration of PC3-B2 cells. Treatment of PC3-B cells with DL-propargylglycine, a CTH enzymatic inhibitor, reduced their reduced invasiveness but not migration, suggesting that CTH enzyme activity and H₂S are only required for invasiveness, whereas CTH protein is required for both. Previously, Snyder's group had reported that H₂S-mediated sulfhydration of Cys38 in the p65 subunit leads to NF- κ B activation and increased cell survival. To investigate whether CTH/H₂S-elicited phenotypes are driven through NF- κ B activation, the authors demonstrated that sulfhydration of p65 was increased in the bone marrow-derived PC3 cells, and that this was reduced by CTH knockdown. They also found that CTH levels correlated with secretion of IL-1 β , a known target gene for NF- κ B, and also increased nuclear localization of p65, suggesting that CTH activity increases NF- κ B activity by inducing its nuclear translocation. Treatment with either NaHS or GYY4137, both H₂S generators, induced PC3 cell invasion but not migration; moreover, NaHS treatment overrode the loss of PC3 cell invasion when CTH was depleted. NaHS treatment increased both p65 sulfhydration and IL-1 β secretion, and the SN50 and GNZ NF- κ B inhibitors reduced NaHS-induced cell invasion. Next, the authors used CRISPR/Cas9 to generate p65 knockout PC3 cells, and then transiently re-expressed WT or C38S mutant p65 in these cells. They found that loss of p65 reduced NaHS-induced invasion, with WT but not C38S p65 re-expression reversing this effect. They went on to show that cells depleted of CTH by stable expression of an shRNA exhibited decreased orthotopic tumor growth and the occurrence of paraaortic lymph node metastases. PC3 cells were detected in the bone marrow of 2/10 animals with orthotopic tumors expressing an shRNA control but none with tumors derived from cells shRNA depleted for CTH. They also found a reduced density of vascular and lymphatic vessels in the tumors of mice with CTH-depleted PC3 cells, suggesting that CTH/H₂S signaling is important for tumor angiogenesis and lymphangiogenesis. Finally, the authors examined CTH protein expression levels in human prostate cancer tumor samples sorting by IHC, and observed a correlation between high CTH levels and worse survival prognosis.

The finding that elevated levels of CTH/CSE protein and its ability to generate the H₂S second messenger lead to p65 sulfhydration and activation of NF- κ B driven expression of IL-1 β and many other genes in PC3 PCa cells is interesting. Based on the data, the tumorigenic effects of CTH overexpression seem quite likely to be due to H₂S production (although CTH does carry out other metabolic reactions that do not generate H₂S). However, while the downstream consequences of elevated H₂S may depend in part on NF- κ B activation and IL-1 β induction, other H₂S targets may also play a role in the tumorigenic properties of prostate cancer cells including migration. The correlation between CTH protein levels and tumorigenesis/metastasis in human PCa patients is interesting and is consistent with the mouse data. In summary, the finding of a possible role for CTH in bone marrow metastasis in prostate cancer is potentially an advance in understanding how bone marrow metastasis arise in prostate cancer, and this could ultimately become a therapeutic target. However, there are a number of issues that need to be addressed.

1. One main issue with these studies is that PC3 cells themselves were derived from a prostate cancer bone metastasis, and therefore it is not clear how valid it is to compare the parental PC3 cells to PC3 cells that migrated to bone from an orthotopically grown PC3 tumor in mice, especially since it is not clear that the mechanism through which human bone metastases originating from a human PCa cell xenograft occurs in immunocompromised mice will be the same as the way bone that metastases arise in humans from a primary PCa. In this regard, the description of how the PC3-T and PC3-B cells were derived needs to be expanded - one presumes that the seven pairs of cell lines correspond to seven different mice, but from the authors' cited paper (Lin et al. (2012)) it is not clear how the PC3-T and PC3-B cells were generated. Are these clones or pools of cells, and how many PC3 cells could be sorted from bone marrow of a mouse with an orthotopic PC3 tumor (was their RNA content analyzed directly after isolation, or after growing out - in the legend to Figure S1 it states that the cells were grown in bone marrow)? If the PC3-B cells are reinoculated either orthotopically or via the tail vein do they exhibit increased migration to the bone marrow and establishment of true metastatic tumors by comparison with PC3-T cells - from what is described, it is not clear whether the cells isolated from bone marrow of mice bearing a primary xenograft were actually growing as metastatic tumors or were simply resident in the bone marrow.

2. The evidence for CTH-generated H₂S, and H₂S-mediated, sulfhydration-dependent NF- κ B activation is reasonable, but they did not show that the level of p65 sulfhydration changed in response to over- or under-expression of CTH, nor was there any attempt to determine the stoichiometry of p65. Can the authors obtain an estimate of the p65 C38 sulfhydration stoichiometry (can the biotin switch assay be used to measure stoichiometry?), and is the stoichiometry high enough after NaHS treatment to explain nuclear translocation of 40% of the p65 population. Snyder's group did not mention any effect of p65 sulfhydration on nuclear translocation, but instead reported that sulfhydration increased p65 binding to NF- κ B response elements in vitro. More importantly, have the authors elucidated a mechanism for how sulfhydration of C38 leads to p65 nuclear translocation?

3. How does the CTH protein promote cell migration in the absence of its enzymatic activity and increased H₂S?

4. It would be reassuring if the authors were able to obtain similar results with another human PCa cell line, and ideally one that that was derived from a primary PCa tumor (there are such lines), where they could show that CTH overexpression induces metastases rather than starting with a cell line that already has known metastatic potential.

Points:

1. Figure 2: These experiments require re-expression of an siRNA-resistant CTH experiments to establish the specificity of the single CTH siRNA used here for CTH knockdown. The top panel is not described in the legend - what are these bands?

2. Page 12/Figure 3: Did IL-1 β treatment reverse the decrease in VEGF and MMP-13 expression?

3. Page 13/Figure 4A: It is surprising that there was no invasion dose response with increasing levels of NaHS; the authors should determine whether the level of p65 sulfhydration is the same at all these doses of NaHS.

4. Page 14/Figure 4C: Knockdown of CTH clearly caused a defect in IL-1 β -induced p65 nuclear translocation in PC3 cells, but it is unclear how p65 sulfhydration would promote nuclear translocation, and the authors did not show that NaHS treatment reverses the CTH knockdown defect. In the text it says that NaHS stimulated p65 sulfhydration in the presence of IL-1 β , but the legend to this panel says nothing about IL-1 β . The results in Figure S5 demonstrate that p65 C38 is required for IL-1 β induced p65 translocation, presumably because it is sulfhydrated. Does this mean that IL-1 β treatment itself induces p65 sulfhydration and, if so, how? Does it activate CTH?

5. Figure S4D: Even though the authors say that the level of nuclear p65 was increased in 40% of H₂S-treated cells, the actual p65 signal per nucleus of H₂S-induced nuclear translocation of p65 is much less impressive than the level of nuclear localization induced by IL-1 β in Figure 3C/D. This suggests that IL-1 β activates additional pathways for p65 nuclear translocation.

6. Figure 4H: Which PC3 cell type was used for the knockdowns - i.e. PC3-T or PC3-B or the parental PC3 cells? The authors need to use p65 knockout PC3 cells stably re-expressing WT and C38S p65 for these experiments, rather than transiently expressing cells.

7. Figure 5: Which PC3 cells were used for these experiments? The authors need to repeat the tumor xenograft experiments with the p65 knockout PC3 cells stably re-expressing WT or C38S p65 in order to directly establish a role for p65 sulfhydration and activation of NF- κ B and IL-1 β expression in both orthotopic tumor growth and bone metastasis.

Referee #2:

This is an interesting article that investigates the role of CTH enzyme in the metastasis and cell invasion of prostatic cancer cells.

The authors showed that CTH produces H₂S that subsequently sulfhydrates the cysteine part of NF- κ B P65 subunit. The activated NF- κ B transcription factor translocates into the cell nucleus and induces the production of IL-1 β .

They showed that the CTH expression is higher in metastatic cancer cells and therefore these cells produced more H₂S.

Next, they showed that induced overexpression or knocking down of CTH in prostatic cancer cells upregulates or downregulates the H₂S production respectively. Interestingly there is a positive correlation between CTH expression and cells invasion and migration. They found that the CTH knocked-down cells produce less IL-1 β and were less invasive and this phenomenon was inverted by adding IL-1 β .

Immunohistochemistry showed the reduced translocation of NF- κ B into the nucleus in knocked down cells and subsequent decreased production of IL-1 β . Interestingly the expression of VEGF and MMP-13 was correlated to IL-1 β production.

Adding exogenous H₂S donors rescued the cell migration, invasion, and expression of IL-1 β , VEGF and MMP-13 in knocked-down cells.

These in-vitro data was confirmed in-vivo in nude mice model. CTH knocked-down prostatic cancer cell line in comparison to control generated less metastasis and smaller tumors. Moreover, the downregulated expression of CD3 and LYVE-1 showed the inhibition of angiogenesis in the CTH knocked-down tumors.

Finally, they successfully showed there is a correlation between CTH expression and stage of prostate cancer in human patient samples.

The experiments are well designed, and the results are clear and convincing.

My suggestion is to improve the introduction by adding the role of 3MST in H₂S production. 3-Mercaptopyruvate sulfurtransferase (3MST) produces H₂S from 3-mercaptopyruvate (3MP), which is generated by cysteine aminotransferase (CAT) and D-amino acid oxidase (DAO) from L-cysteine and D-cysteine, respectively.

I also recommend adding more details in figures legends, especially in supplementary figures.

In statistics, in each figure legend, please clarify that how many times each experiment was repeated with similar results.

Referee #3:

In this manuscript, Wang et al describe a previously unknown mechanism involving upregulated cystathionine gamma-lyase (CTH) expression and its product hydrogen sulfide in prostate cancer progression and metastasis. The authors demonstrate that gene and protein expression of CTH, as well as its product (hydrogen sulfide) are elevated in bone metastatic PC3 cells compared to primary tumor-derived cells. This is shown using multiple cell lines for validation. Genetic downregulation of CTH in PC3 cells results in decreased migration and invasion via NF- κ B and IL-1 signaling. The data presented for these experiments are sufficient to support these statements. The authors propose that the underlining mechanism for this observation involves phosphorylation of NF- κ B at the p65 subunit causing the enhanced cell invasion and migration. Elegant loss and gain-of-function experiments provide sufficient experimental evidence to support this phenotype. Loss of CTH expression caused reduced tumor growth in vivo as well as bone metastasis and reduced lymph node numbers. Finally, CTH expression was increased in primary prostate cancer tumors compared to adjacent normal tissue. TCGA data showed that CTH expression was significantly increased in prostate cancer tissues.

In summary, the findings described in this manuscript are strongly documented with physiological relevance and of wide interest. Functional data that are presented here are of very high quality and appropriate control samples have been used where needed. The statistical methods used are the

appropriate for the comparisons performed. Use of mouse models and human data further strengthen these findings.

Minor weaknesses:

- In figure 3D the quantification of the immunofluorescence images show a significant translocation of p65 from the nucleus (siCTH) to the cytoplasm (suCTH+IL β). The images presented at 3D however do not represent that strong of a difference. Nuclear/cytoplasmic isolation of p65 and quantification would provide more compelling evidence of NF- κ B activation.
- Grammar errors throughout the manuscript, mostly in the introduction section.

1st Revision - authors' response

17 July 2019

Referee #1:

1. One main issue with these studies is that PC3 cells themselves were derived from a prostate cancer bone metastasis, and therefore it is not clear how valid it is to compare the parental PC3 cells to PC3 cells that migrated to bone from an orthotopically grown PC3 tumor in mice, especially since it is not clear that the mechanism through which human bone metastases originating from a human PCa cell xenograft occurs in immunocompromised mice will be the same as the way bone that metastases arise in humans from a primary PCa. In this regard, the description of how the PC3-T and PC3-B cells were derived needs to be expanded - one presumes that the seven pairs of cell lines correspond to seven different mice, but from the authors' cited paper (Lin et al. (2012)) it is not clear how the PC3-T and PC3-B cells were generated. Are these clones or pools of cells, and how many PC3 cells could be sorted from bone marrow of a mouse with an orthotopic PC3 tumor (was their RNA content analyzed directly after isolation, or after growing out - in the legend to Figure S1 it states that the cells were grown in bone marrow)? If the PC3-B cells are reinoculated either orthotopically or via the tail vein do they exhibit increased migration to the bone marrow and establishment of true metastatic tumors by comparison with PC3-T cells - from what is described, it is not clear whether the cells isolated from bone marrow of mice bearing a primary xenograft were actually growing as metastatic tumors or were simply resident in the bone marrow.

We appreciate the reviewer's in-depth thought and valuable advice. Indeed PC3 cells were originally derived from bone metastasis, so they exhibit higher metastatic potential than other human prostate cancer lines. Therefore, PC3 cells have been well characterized as the metastasis mouse model in most prostate cancer studies. Unfortunately, we weren't able to count the number of PC3 cells obtained from bone marrow from each tumor-bearing mouse. Those cells were isolated from bone marrow and expanded on the petri dish for a few weeks to achieve sufficient cell population for biochemical analysis including gene expression profiles and functional assays. To clarify how these cell lines were generated, we have added more detailed description in the result section. The content is excerpted below:

“To elucidate the differential expression of genes during PC metastasis, we established paired PC3-derived cell lines isolated and established from tumor cells in the bone marrow and from primary tumors, respectively, using xenograft orthotopic implantation mouse model (Lin, Gong et al., 2012). Briefly, PC3 cells were injected into the ventral portion of the prostate in nude mice and allowed 40 days for the growth of the primary tumor and potential distant metastasis. We then cultured tumor cells isolated from the primary prostate tumors as the “tumor-derived PC3 cells” or T-lines, as well as tumor cells from thigh bone marrows to obtain the “bone-metastatic PC3 cells” or B-lines. Seven paired PC3 B- and T- lines were thus established from bone marrows and primary tumors, respectively, from seven PC3 tumor-bearing nude mice.”

In addition, to determine whether B-lines exhibited higher metastatic potential than T lines, we re-inoculated PC3-B2, B3 and -T2, T3 cells orthotopically into the nude mice prostates. Our data indicated that PC3-B2, B3 exhibited higher metastatic potential than PC3-T2, T3 cells (Fig. 1E, F, Appendix Fig S2), indicating that PC3-B2, B3 cells gained increased migratory ability to disseminate to distant organs such as paraaortic lymph nodes and bone marrow and grew into micro or macro metastatic tumors, whereas PC3-T2, T3 cells had much less abilities to do so.

2. The evidence for CTH-generated H₂S, and H₂S-mediated, sulfhydration-dependent NF- κ B

activation is reasonable, but they did not show that the level of p65 sulfhydrylation changed in response to over- or under-expression of CTH, nor was there any attempt to determine the stoichiometry of p65. Can the authors obtain an estimate of the p65 C38 sulfhydrylation stoichiometry (can the biotin switch assay be used to measure stoichiometry?), and is the stoichiometry high enough after NaHS treatment to explain nuclear translocation of 40% of the p65 population. Snyder's group did not mention any effect of p65 sulfhydrylation on nuclear translocation but instead reported that sulfhydrylation increased p65 binding to NF- κ B response elements in vitro. More importantly, have the authors elucidated a mechanism for how sulfhydrylation of C38 leads to p65 nuclear translocation?

We have checked the levels of p65 sulfhydrylation in PC3-T2, B2 lines, PC3-T3, B3 lines, and PC3-B3 line with CTH knockdown (Fig EV3A). We observed increased sulfhydrylation on p65 in the PC3-B lines, as compared to the PC3-T lines, and p65 sulfhydrylation was decreased upon CTH knockdown. The p65 sulfhydrylation levels in PC3-T/B paired lines and upon different treatments are shown by histograms (Fig EV3A). We also determined the levels of p65 sulfhydrylation after treatment with different amount of NaHS (Fig EV4C). We observed induction of p65 nuclear translocation in the presence of NaHS (1 μ M; Fig EV4D), and rescue of reduced cell invasion and p65 nuclear translocation by CTH knockdown (Fig 4B, Appendix Fig. S3). Mutation of p65 on cysteine 38 resulted in the absence of p65 sulfhydrylation (Fig EV4H), indicating that cysteine 38 is the only cysteine on p65 for sulfhydrylation.

So far our data indicate that p65 sulfhydrylation can facilitate nuclear translocation, and mutation on cysteine 38 resulted in the absence of sulfhydrylation (Fig EV4H) and impaired nuclear translocation of p65 (Appendix Fig S4). We also checked whether p65^{C38S} mutation affected I κ B α degradation upon IL-1 β stimulation and found that I κ B α degradation is not affected in cells expressing p65^{C38S} mutant (data not shown). We are currently working on the identification of proteins that interact with p65 after its sulfhydrylation by immunoprecipitation. Our preliminary results showed that p65^{C38S} failed to interact with importin α and β in the presence of IL-1 β (data not shown). However, these data still need to be repeated to draw the conclusion. Since importin α and β are key molecules for NF- κ B nuclear transport, we believe that sulfhydrylation of p65 may function in the interaction between p65 and importin α/β to facilitate nuclear translocation of NF- κ B. We are currently working on the detailed mechanism of how p65 sulfhydrylation facilitates its nuclear translocation.

3. How does the CTH protein promote cell migration in the absence of its enzymatic activity and increased H₂S?

We are very curious about this phenomenon. To address this question, we generated an enzyme-dead CTH mutant, Q240E. According to Banerjee's study in 2008, the CTH Q240E mutation weakens its enzymatic activity by 70 folds (Zhu, Lin, et al., 2008). Here we found that overexpression of the mutant could still promote prostate cancer cell migration, but much less so in promoting invasion in comparison with the cells expressing wild type CTH. Our data showed that CTH could induce MMP-13 (Fig 3G) via H₂S-mediated signaling pathway (Fig 4E), which is known to promote invasion. The enzymatic activity of CTH may be involved in MMP-13 induction, and thus impairment of enzymatic activity would lead to substantial loss of invasion promoting ability. Due to the fact that the enzymatic activity of CTH is not completely abolished in this Q240E mutant, there existed residual invasion promoting ability in the cells expressing this mutant. Overall, our data indicate that CTH promotes cell migration through an enzyme-independent pathway. In the future, we will try to identify proteins interacting with this mutant to decipher this unknown mechanism of how CTH promotes cell migration.

4. It would be reassuring if the authors were able to obtain similar results with another human PCa cell line, and ideally one that that was derived from a primary PCa tumor (there are such lines), where they could show that CTH overexpression induces metastases rather than starting with a cell line that already has known metastatic potential.

Thank you for the valuable advice. To examine whether CTH plays important roles during prostate cancer progression and metastasis, we generated DU145- and LNCaP- derived cell lines stably over-expressing CTH. Expression of CTH was significantly lower in both cell lines, as compared to PC3 cells (Fig EV1C). The LNCaP line was originally derived from left supraclavicular lymph node metastases, and DU145 line was originally derived from brain metastasis (Stone, Mickey, et al.,

1978). Neither line was derived from bone metastasis. We implanted these two lines with or without CTH overexpression to assess whether CTH promoted distant metastasis in the orthotopic implantation mouse model. Overexpression of CTH in DU145 promoted tumor growth and increased the incidence of paraaortic lymph nodes metastases (Fig EV5). In LNCaP cells overexpressing CTH, tumor growth was significantly increased. However, no lymph nodes or bone metastases were observed (Data not shown), implying LNCaP cells had much lower metastatic potential, and CTH alone is not sufficient to promote their metastasis in this model. The above details have been added to the results section. The content is excerpted below:

“To investigate whether CTH is required for PC progression, we stably expressed CTH in DU145 cell line. DU145 line was originally derived from brain, and not bone, metastasis (Stone et al., 1978), and the expression level of CTH was much lower in DU145 cells, as compared to other prostate cancer lines (Fig EV1C). The control and CTH stable overexpressing lines were established (Fig EV5A) and orthotopically injected into the prostate capsules of nude mice. In the CTH overexpressing group, the mouse weight was slightly decreased as compared to the control group (Fig EV5B), and tumor size was significantly increased in comparison with the control group (Fig EV5C, E). The incidence of paraaortic lymph node metastasis was significantly increased in the CTH group as compared to the control group (Fig EV5D, F, G). However, bone metastasis was not detected in either group under our experimental duration. Overall, our data suggest that CTH is capable of promoting tumor growth, as well as lymph node and bone metastasis in certain PC lines such as PC3. However, CTH may be required, but not sufficient, for promoting bone metastasis.”

Points:

1. Figure 2: These experiments require re-expression of a siRNA-resistant CTH experiment to establish the specificity of the single CTH siRNA used here for CTH knockdown. The top panel is not described in the legend - what are these bands?

Thank you for the valuable advice. To ensure the specificity of our siRNA. We use two different siRNAs targeting CTH to perform migration and invasion experiments. Reduced migration and invasion by CTH knockdown were rescued in cells co-transfected with CTH overexpression plasmid (Fig EV2A, B). For Fig 2B, more details are added to the description. The contents are excerpted below:

"(B) Top: The H₂S production capacity of cell lysates from PC3-T2 cells overexpressing CTH or CTH knockdown PC3-B2 cells. Lead acetate-soaked paper strips show a PbS brown stain as a result of reaction with H₂S. Bottom: The level of H₂S production was quantified by densitometry, and the histograms represent the means \pm SD (n=3 independent experiments)"

2. Page 12/Figure 3: Did IL-1 β treatment reverse the decrease in VEGF and MMP-13 expression?

This experiment has been done, and results are shown in Fig 3F, G. IL-1 β treatment could partially reverse VEGF and MMP-13 expression by CTH knockdown.

3. Page 13/Figure 4A: It is surprising that there was no invasion dose response with increasing levels of NaHS; the authors should determine whether the level of p65 sulphydration is the same at all these doses of NaHS.

We have tried to determine whether there are dose-dependent effects on p65 sulphydration by the biotin-switch assay. Our data revealed that the levels of p65 sulphydration were similar from 10nM-10 μ M of NaHS (Fig. EV4C), and these doses of NaHS induced the highest invasion ability as compared to other concentrations we tested (Fig. 4A). Interestingly, the level of p65 sulphydration decreased upon dosage higher than 100 μ M (Fig. EV4C), implying that certain unknown inhibitory mechanism may exist. We also noticed that there was no dose-dependent response in IL-1 β protein production upon NaHS treatment (10nM-10 μ M) (Fig 4F), which indirectly reflects the similar phenomenon of cell invasion ability promoted by NaHS from 10nM to 10 μ M.

4. Page 14/Figure 4C: Knockdown of CTH clearly caused a defect in IL-1 β -induced p65 nuclear translocation in PC3 cells, but it is unclear how p65 sulphydration would promote nuclear translocation, and the authors did not show that NaHS treatment reverses the CTH knockdown defect. In the text, it says that NaHS stimulated p65 sulphydration in the presence of IL-1 β , but the legend to this panel says nothing about IL-1 β . The results in Figure S5 demonstrate that p65 C38 is

required for IL-1 β induced p65 translocation, presumably because it is sulfhydrated. Does this mean that IL-1 β treatment itself induces p65 sulfhydration and, if so, how? Does it activate CTH?

First, we would like to apologize for the typo of this sentence "NaHS stimulated p65 sulfhydration in the presence of IL-1 β ", which we did not treat IL-1 β in Fig 4C. To further address reviewer's question, we showed that NaHS treatment can reverse the suppression of cell invasion (Fig 4B) and p65 nuclear translocation (Appendix Fig S3) by CTH knockdown, indicating that sulfhydration of p65 can facilitate p65 nuclear translocation. We also checked whether IL-1 β treatment itself can induce CTH expression. After 24hr treatment of IL-1 β in CTH knockdown cell, the level of CTH remained similar (Fig EV3D) and so was the level of p65 sulfhydration (Fig EV3E), indicating that IL-1 β is not involved in CTH induction or p65 sulfhydration. Our working model is that CTH level is elevated in bone-metastatic prostate cancer cells, and thus it facilitates IL-1 β -induced NF- κ B activation, resulting in the induction of MMP13 and VEGF-A to promote cell invasion and metastasis (Appendix Fig S1).

5. Figure S4D: Even though the authors say that the level of nuclear p65 was increased in 40% of H₂S-treated cells, the actual p65 signal per nucleus of H₂S-induced nuclear translocation of p65 is much less impressive than the level of nuclear localization induced by IL-1 β in Figure 3C/D. This suggests that IL-1 β activates additional pathways for p65 nuclear translocation.

We agree with the reviewer's comment that IL-1 β may activate additional pathways for p65 nuclear translocation. To discuss how p65 sulfhydration affects nuclear translocation, we addressed this issue in the Discussion Section. The content is excerpted below:

" NF- κ B activation requires translocation of NF- κ B subunits, p65 and p50, from cytosol to the nucleus (Sun & Andersson, 2002, Zhang, Lenardo et al., 2017). Nuclear translocation of the NF- κ B is initiated by the signal-induced degradation of I κ B proteins through activation I κ B kinase (IKK). The degradation of I κ B thus releases NF- κ B to translocate into the nucleus and activate gene transcriptions (Napetschnig & Wu, 2013). Here our data showed that blocking p65 sulfhydration resulted in the attenuation of p65 nuclear translocation induced by IL-1 β (Fig 3D, Appendix Fig S4), suggesting sulfhydration of p65 might be involved in the nuclear import of the p65 subunit. We also noticed that treatment with H₂S alone only induced modest nuclear translocation of p65 (Fig EV4D), and this induction is incomparable to the level of IL-1 β -induced nuclear translocation of p65 (Fig 3D). Based on these observations, we believe that p65 sulfhydration by H₂S is not enough to stimulate the p65 nuclear translocation since NF- κ B complex may still interact with the inhibitory protein I κ B. Additional signals, such as IL-1 β , is required to activate IKK through phosphorylation, resulting in the degradation of I κ B to release p65. The p65 sulfhydration may be required for the interaction between p65 and nuclear transport proteins to facilitate nuclear import. More research is needed to determine the exact role of p65 sulfhydration in regulating NF- κ B activity."

6. Figure 4H: Which PC3 cell type was used for the knockdowns - i.e. PC3-T or PC3-B or the parental PC3 cells? The authors need to use p65 knockout PC3 cells stably re-expressing WT and C38S p65 for these experiments, rather than transiently expressing cells.

We use parental PC3 cells (not PC3-T or PC3-B cells) for the knockdown experiments. Our p65 knockout PC3 cells were Puromycin and Blasticidin resistant resulting from Cas9 and sgRNA plasmid selection. Therefore, in order to re-expressing WT or C38S mutant p65 in the p65 knockout PC3 cells, we sub-cloned WT or mutant p65 gene into pCDNA3.1, which contains G418 resistance gene. We then tried to establish PC3 cells stably expressing p65^{wild-type} or p65^{C38S}. However, although the knock-out PC3 cells could transiently express p65^{wild-type} or p65^{C38S}, the cells somehow lost the expression vector after one month of culturing and drug selection. We first considered the possibility that the sgRNA and Cas9 might still be present inside the cells and target the excision of the expression vector. So we proceeded to make p65^{wild-type} and p65^{C38S} in-frame mutation to replace sgRNA recognizing nucleotides. Unfortunately, even by transfecting with those mutated forms, there were still no p65^{wild-type} or p65^{C38S} expression in cells grown after drug selection. So far, we have been trying every possible attempt to establish those stable cell lines.

On the other hand, although stable lines expressing p65 (wild-type or C38S) are more suitable for the long term *in vivo* experiments, transient expression could still provide sufficient information in terms of cell invasion and nuclear translocation without the complication of long term culturing of the stable expressing lines. Our data indicated that transient re-expression of p65^{wild-type}, but not

p65^{C38S} mutant, restored invasion abilities (Fig 4H), as well as nuclear translocation of p65 in the presence of IL-1 β (Appendix Fig S4A, B).

7. Figure 5: Which PC3 cells were used for these experiments? The authors need to repeat the tumor xenograft experiments with the p65 knockout PC3 cells stably re-expressing WT or C38S p65 in order to directly establish a role for p65 sulphydration and activation of NF- κ B and IL-1 β expression in both orthotopic tumor growth and bone metastasis.

For Figure 5, we used PC3 cells (not PC3-B2 or PC3-B3 cells) with shCon or shCTH expression for xenograft mouse experiment. To investigate whether p65 sulphydration plays a role in cancer metastasis, we transiently re-expressed p65^{wild-type} or p65^{C38S} in PC3 cells with p65 knock-out, inoculated 1x10⁶ cells into the bloodstream via tail-vein injection, and harvested lung to examine lung metastasis by H&E staining after 7 days of inoculation. The p65 knockout PC3 cells with this p65^{C38S} mutant re-expression showed a decreased number of lung metastatic nodules (Appendix Fig S5A, B) and reduced percentage of lung metastatic area (Appendix Fig S5C), as compared to the p65^{wild-type} re-expression group, suggesting that p65 sulphydration could be directly involved in the promotion of cancer metastasis. However, we still need more evidence to draw firm conclusions that p65 sulphydration at cysteine 38 is the key factor required for PC tumor growth and bone metastasis. A better way to answer it is to perform genome editing of p65 replacing cysteine 38 to serine, and we are currently working on designing the CRISPR-based knock-in strategy to establish this knock-in PC3 cell line for future *in vivo* investigation.

Referee #2:

My suggestion is to improve the introduction by adding the role of 3MST in H₂S production. 3-Mercaptopyruvate sulfurtransferase (3MST) produces H₂S from 3-mercaptopyruvate (3MP), which is generated by cysteine aminotransferase (CAT) and D-amino acid oxidase (DAO) from L-cysteine and D-cysteine, respectively.

Thanks to the reviewer's suggestion. We have added the role of 3MST into the introduction section. The content is excerpted below:

"H₂S produces endogenously as a metabolite by cystathionine β -synthase (CBS), cystathionine γ -lyase (CTH), and 3-mercaptopyruvate sulfurtransferase (3MST). CBS catalyzes H₂S by beta-replacement reaction with cysteine (Chen, Jhee et al., 2004), while CTH produces H₂S from cysteine and water through alpha & beta-elimination. In higher concentration of homocysteine, CTH also generates homolanthionine and H₂S from two homocysteines by gamma-replacement reactions (Kabil, Vitvitsky et al., 2011). Other than CBS and CTH, 3MST produces H₂S by processing 3-mercaptopyruvate (3MP), a metabolite catalyzed from L- and D-cysteine by cysteine aminotransferase (CAT) and D-amino acid oxidase (DAO), respectively (Cooper, 1983, Meister, Fraser et al., 1954, Shibuya, Mikami et al., 2009, Shibuya, Tanaka et al., 2009). Endogenous H₂S derived from these three enzymes plays promoting roles on tumor growth in a variety of different cancer types through induction of angiogenesis, regulation of mitochondrial bioenergetics, acceleration of cell cycle progression and anti-apoptosis (Wu, Si et al., 2015). Among them, CBS is found to be upregulated and in thyroid (Turbat-Herrera, Kilpatrick et al., 2018), ovarian (Bhattacharyya, Saha et al., 2013), and colon cancers (Phillips, Zatarain et al., 2017, Untereiner, Pavlidou et al., 2018), while CTH promotes cell growth in breast cancer cells (Wang, Shi et al., 2019) and astrocytoma cells (Jurkowska & Wrobel, 2018). Increased expressions of 3MST were observed in colon, lung adenocarcinoma, urothelial cell carcinoma, and oral carcinomas, implying 3MST may play a role in drug resistance and oxidative damage during cancer progression (Augsburger & Szabo, 2018). Notably, whether H₂S affects cancer metastasis remains unclear."

I also recommend adding more details in figures legends, especially in supplementary figures.

Thank you for the advice. We have added more details in figure legends, especially in supplementary figures.

In statistics, in each figure legend, please clarify that how many times each experiment was repeated with similar results.

We have included the number of times each experiment was repeated in all figures and material and methods.

Referee #3:

Minor weaknesses:

- In figure 3D the quantification of the immunofluorescence images show a significant translocation of p65 from the nucleus (siCTH) to the cytoplasm (suCTH+IL β). The images presented at 3D however do not represent that strong of a difference. Nuclear/cytoplasmic isolation of p65 and quantification would provide more compelling evidence of NF- κ B activation.

We appreciate the advice. We performed the nuclear/cytoplasmic fractionation to quantify IL-1 β induced p65 nuclear translocation (Figure EV3B, C) and further confirmed our findings. On the other hand, for Fig 3D, we have replaced the original images to more representative images now.

- Grammar errors throughout the manuscript, mostly in the introduction section.

Thank you for the advice. To improve our writing, we have sent the manuscript to the language editing company for professional English editing. The certificate of English editing is provided as below:

Reference:

- Augsburger F, Szabo C (2018) Potential role of the 3-mercaptopyruvate sulfurtransferase (3-MST)-hydrogen sulfide (H₂S) pathway in cancer cells. *Pharmacol Res*
- Bhattacharyya S, Saha S, Giri K, Lanza IR, Nair KS, Jennings NB, Rodriguez-Aguayo C, Lopez-Berestein G, Basal E, Weaver AL, Visscher DW, Cliby W, Sood AK, Bhattacharya R, Mukherjee P (2013) Cystathionine beta-synthase (CBS) contributes to advanced ovarian cancer progression and drug resistance. *PLoS One* 8: e79167
- Chen X, Jhee KH, Kruger WD (2004) Production of the neuromodulator H₂S by cystathionine beta-synthase via the condensation of cysteine and homocysteine. *J Biol Chem* 279: 52082-6
- Cooper AJ (1983) Biochemistry of sulfur-containing amino acids. *Annu Rev Biochem* 52: 187-222
- Jurkowska H, Wrobel M (2018) Cystathionine Promotes the Proliferation of Human Astrocytoma U373 Cells. *Anticancer Res* 38: 3501-3505
- Kabil O, Vitvitsky V, Xie P, Banerjee R (2011) The quantitative significance of the transsulfuration enzymes for H₂S production in murine tissues. *Antioxid Redox Signal* 15: 363-72
- Lin KT, Gong J, Li CF, Jang TH, Chen WL, Chen HJ, Wang LH (2012) Vav3-rac1 signaling regulates prostate cancer metastasis with elevated Vav3 expression correlating with prostate cancer progression and posttreatment recurrence. *Cancer Res* 72: 3000-9
- Meister A, Fraser PE, Tice SV (1954) Enzymatic desulfuration of beta-mercaptopyruvate to pyruvate. *J Biol Chem* 206: 561-75
- Napetschnig J, Wu H (2013) Molecular basis of NF-kappaB signaling. *Annu Rev Biophys* 42: 443-68
- Phillips CM, Zatarain JR, Nicholls ME, Porter C, Widen SG, Thanki K, Johnson P, Jawad MU, Moyer MP, Randall JW, Hellmich JL, Maskey M, Qiu S, Wood TG, Druzhyna N, Szczesny B, Modis K, Szabo C, Chao C, Hellmich MR (2017) Upregulation of Cystathionine-beta-Synthase in Colonic Epithelia Reprograms Metabolism and Promotes Carcinogenesis. *Cancer Res* 77: 5741-5754
- Shibuya N, Mikami Y, Kimura Y, Nagahara N, Kimura H (2009) Vascular endothelium expresses 3-mercaptopyruvate sulfurtransferase and produces hydrogen sulfide. *J Biochem* 146: 623-6
- Shibuya N, Tanaka M, Yoshida M, Ogasawara Y, Togawa T, Ishii K, Kimura H (2009) 3-Mercaptopyruvate sulfurtransferase produces hydrogen sulfide and bound sulfane sulfur in the brain. *Antioxid Redox Signal* 11: 703-14
- Stone KR, Mickey DD, Wunderli H, Mickey GH, Paulson DF (1978) Isolation of a human

- prostate carcinoma cell line (DU 145). *Int J Cancer* 21: 274-81
- Sun Z, Andersson R (2002) NF-kappaB activation and inhibition: a review. *Shock* 18: 99-106
- Turbat-Herrera EA, Kilpatrick MJ, Chen J, Meram AT, Cotelingam J, Ghali G, Keivilian CG, Coppola D, Shackelford RE (2018) Cystathionine beta-Synthase Is Increased in Thyroid Malignancies. *Anticancer Res* 38: 6085-6090
- Untereiner AA, Pavlidou A, Druzhyna N, Papapetropoulos A, Hellmich MR, Szabo C (2018) Drug resistance induces the upregulation of H2S-producing enzymes in HCT116 colon cancer cells. *Biochem Pharmacol* 149: 174-185
- Wang L, Shi H, Zhang X, Zhang X, Liu Y, Kang W, Shi X, Wang T (2019) I157172, a novel inhibitor of cystathionine gamma-lyase, inhibits growth and migration of breast cancer cells via SIRT1-mediated deacetylation of STAT3. *Oncol Rep* 41: 427-436
- Wu D, Si W, Wang M, Lv S, Ji A, Li Y (2015) Hydrogen sulfide in cancer: Friend or foe? *Nitric Oxide* 50: 38-45
- Zhang Q, Lenardo MJ, Baltimore D (2017) 30 Years of NF-kappaB: A Blossoming of Relevance to Human Pathobiology. *Cell* 168: 37-57
- Zhu W, Lin A, Banerjee R (2008) Kinetic properties of polymorphic variants and pathogenic mutants in human cystathionine gamma-lyase. *Biochemistry* 47: 6226-32

2nd Editorial Decision

5 August 2019

Thank you for the submission of your revised manuscript to our editorial offices. We have now received the reports from the two referees that were asked to re-evaluate your study, you will find below. As you will see, both referees now support the publication of your study in EMBO reports. Referee #1 has a final suggestion I ask you to address in a final revised version of your manuscript. Please try to add Figures S1 (maybe as last main figure), S4 and S5 to the main paper (as part of the main figures, or the EV figures).

As your paper will be published as Article, there is no limitation regarding the manuscript size. Thus, you could increase the number of main figures. Only, for the EV figures, there is a limitation of five. Presently, the figures are not that crowded, thus I think it would be possible to add the data in figures S4 and S5 to the existing figures as new panels, and feature the summary in Figure S1 as new Figure 6.

Further, I have these editorial requests:

- Please provide the abstract written in present tense.
- Please move the financial support section to the acknowledgements.
- Please format the references according to our journal style. See: <http://www.embopress.org/page/journal/14693178/authorguide#referencesformat>
- Thanks a lot for providing the WB source data. This is highly appreciated. Please combine the source data from one figure into one pdf file, and upload these as separate files (for main and EV figures).
- Please indicate in all figure legends (main figures, EV figures and the Appendix) the nature of the n replicates (biological vs. technical).
- For those figures that remain in the Appendix, please add the legends below the figures. This renders the Appendix easier to comprehend for the reader. Please update then the page numbers in the Appendix TOC.
- Please use as title for Appendix Table S2: Clinical data for 469 prostate cancer cases.
- Finally, please find attached a word file of the manuscript text (provided by our publisher) with changes we ask you to include in your final manuscript text, and some queries, we ask you to

address. Please provide your final manuscript file with track changes, in order that we can see the modifications done.

In addition I would need from you:

- a short, two-sentence summary of the manuscript
- two to three bullet points highlighting the key findings of your study
- a schematic summary figure (in jpeg or tiff format with the exact width of 550 pixels and a height of not more than 400 pixels) that can be used as a visual synopsis on our website.

REFEREE REPORTS

Referee #1:

The authors have made a good effort to address the concerns raised by the reviewers, and added a lot of new data. In particular, they have included new experiments with another metastatic PCa cell line, DU145, demonstrating that mice with orthotopically implanted CTH-overexpressing DU145 cells exhibited increased lymph node metastasis, consistent with a role of high CTH levels in PCa metastasis. There are still no insights into how sulfhydrylation of C38 in p65 leads to nuclear import, and in this connection is unfortunate that they were unable to generate stable p65 Δ PC3 cell lines stably re-expressing WT or C38S p65 to carry out long term tumorigenesis/metastasis experiments, and instead have had to rely on p65 Δ PC3 cells transiently expressing WT or C38S p65 and a short term experiment using tail vein injection to assess effects on lung metastatic seeding. They are now using CRISPR/Cas9 technology to make the p65 C38S knock-in mutant in PC3 cells, but this is likely to take some time, especially since they have to make sure that every p65 allele is mutated, and I do not feel it is necessary to wait for these experiments.

In summary, the proposal that cystathionine- γ -lyase upregulation in metastatic PCa cells leads to elevated H₂S resulting from and concomitant sulfhydrylation of NF- κ B p65, which triggers nuclear import in prostate cancer metastasis is novel, and could have therapeutic implications for treatment of metastatic prostate cancer.

Point: Because most people will not bother to download the Appendix, I recommend that the authors find a way to include Figures S4 and S5 from the Appendix in the main paper, since these provide important evidence that p65 C38 is important for nuclear localization and for lung metastasis. And, ideally also add the model in Figure S1 to the main manuscript.

Referee #3:

The authors have addressed all of my initial concerns and added appropriate experimental evidence by performing more in vivo work. Authors proof-read the manuscript using an external English-editing service which significantly improved the manuscript. Revised figures contain appropriate information and are easy to understand. After reviewing the manuscript, I recommend it for publication.

2nd Revision - authors' response

26 July 2019

The authors performed all minor editorial changes.

YOU MUST COMPLETE ALL CELLS WITH A PINK BACKGROUND ↓
PLEASE NOTE THAT THIS CHECKLIST WILL BE PUBLISHED ALONGSIDE YOUR PAPER

Corresponding Author Name: Lu-Hai Wang & Kai-Ti Lin
Journal Submitted to: EMBO Reports
Manuscript Number: EMBOR-2018-45986V3